# Unravelling the role of vacancies in lead halide perovskite through electrical switching of photoluminescence

Cheng Li[1], Antonio Guerrero [2], Sven Huettner[1] & Juan Bisquert [2]

We address the behavior in which a bias voltage can be used to switch on and off the photoluminescence of a planar film of methylammonium lead triiodide perovskite (MAPbI$_3$) semiconductor with lateral symmetric electrodes. It is observed that a dark region advances from the positive electrode at a slow velocity of order of 10 μm s$^{-1}$. Here we explain the existence of the sharp front by a drift of ionic vacancies limited by local saturation, that induce defects and drastically reduce the radiative recombination rate in the film. The model accounts for the time dependence of electrical current due to the ion-induced doping modification, that changes local electron and hole concentration with the drift of vacancies. The analysis of current dependence on time leads to a direct determination of the diffusion coefficient of iodine vacancies and provides detailed information of ionic effects over the electrooptical properties of hybrid perovskite materials.

[1] Department of Chemistry, University of Bayreuth, Universitätstr. 30, 95447 Bayreuth, Germany. [2] Institute of Advanced Materials (INAM), Universitat Jaume I, 12006 Castello, Spain. Correspondence and requests for materials should be addressed to S.H. (email: sven.huettner@uni-bayreuth.de) or to J.B. (email: bisquert@uji.es)

The advent of hybrid perovskite solar cells has given rise to extraordinary photovoltaic performances, causing the rise of a solar energy conversion technology. However, the original physical characteristics of these materials are not yet completely understood, and many significant experimental observations have not been satisfactorily explained so far. Slow time-scale variations of photoluminescence (PL) phenomena have been widely reported since an early stage in the research of perovskite photovoltaics and optoelectronics[1,2]. Sanchez et al.[3] observed either increasing or decreasing PL intensity according to the preparation method of the methylammonium lead triiodide (MAPbI$_3$) layer. Hoke et al.[4] interpreted the observation of light-induced transformations of PL in mixed MAPb(Br$_x$I$_{1-x}$)$_3$ in terms of photoinduced phase segregation. Subsequently, Leitjens et al.[5] observed that an applied electrical field across the perovskite layer can either enhance or suppress the luminescence in lateral interdigitated electrode devices. These findings were described in terms of a simple mechanism common to both fully inorganic and organic semiconductors[6,7], in which photogenerated electrons and holes drift to opposite sides of the device, reducing the bulk recombination rate and hence PL intensity. However, some aspects of the observations cannot be explained by this simple model. Furthermore, it is expected that under an applied field, a massive ion drift will strongly affect the PL characteristics. In fact, the ion displacement has been reported to modify PL properties under different conditions[8,9], and a variety of reversible and irreversible PL transient responses have been obtained under modification of the applied bias magnitude and electrode polarity of laterally contacted MAPbI$_3$ layers[10–14] that still require a general explanation. In this paper, we will present a dynamic transport model, based on the modification of electronic concentrations by the displacement of halogenide vacancies in the perovskites. We show that the model matches well with experimental data of the advancement of a dark edge in a wide variety of material samples, and more importantly, it directly relates the PL effects to the ionic drift controlled by a local saturation effect of defects, allowing the

direct determination of defect densities and ionic diffusion constants.

## Results

**Observation of advancing front characteristics.** Recently, a property was uncovered, revealing striking features of the transient PL in perovskite layers[15]. Using a wide-field PL imaging microscope, it was observed that the PL is progressively suppressed from the positively biased electrode. Here, additional data have been recorded for interdigitated electrodes with a wider channel length of ~150 μm, and they are shown in Fig. 1a. The darkened area in the left of the image forms a sharp front that advances at a slow velocity of ca. 10 μm s$^{-1}$ toward a negative electrode, shown in Supplementary Movie 1. In some cases, the front reaches the negative electrode and PL is completely removed. It can be later restored by biasing the perovskite film in the contrary direction, as shown in Supplementary Movie 2 or letting the system to relax in the absence of bias and dark conditions, although recovery occurs at a much slower rate. These observations were interpreted in terms of ionic redistribution caused by the applied field[11,16], but a quantitative understanding in terms of specific ionic species and a mechanistic suppression of PL could not be achieved. To further understand the mechanism, we carry out the following experiments and establish the electrical model.

In the experiments presented in this work, the PL was optically recorded while the electrical current was measured at a constant bias voltage from less than 1 V for impedance spectroscopy (IS) measurements, ~5 V for MAPbI$_x$Cl$_{3-x}$ and up to 100 V for multiquantum wall (MQW) perovskites. Note that the electrical field in the 150 μm channel is significantly lower than the field experienced in a perovskite layer of 350 nm in photovoltaic devices, as illustrated in Supplementary Table 1. In Fig. 1, we observe that the overall initial PL intensity within the channel decreases during the initial 12 s. In parallel, the measured current shows a rapid decrease from 14 μA to 3.0 μA (Fig. 1g) after a

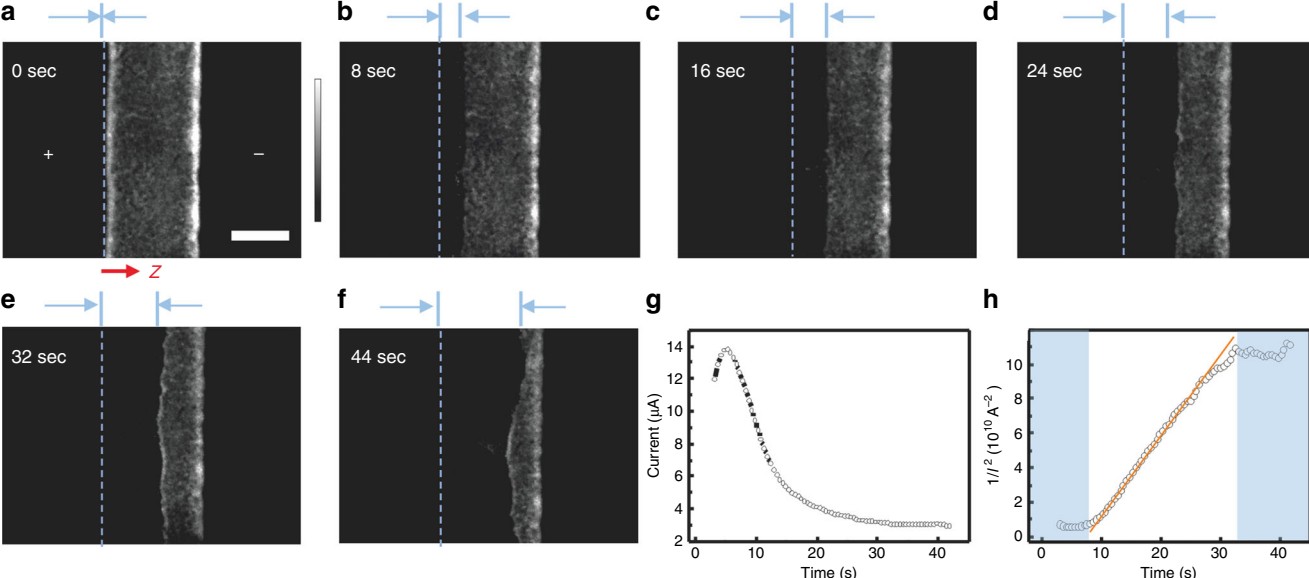

**Fig. 1** Observation of a dark advancing front in a light-soaked perovskite. **a–f** Time-dependent PL images of a perovskite film CH$_3$NH$_3$PbI$_{3-x}$Cl$_x$ under an external electric field (~2 × 10$^4$ V m$^{-1}$). The "+" and '–' signs indicate the polarity of the electrodes. The excitation intensity is ~35 mW cm$^{-1}$ with a wavelength of 440 nm, and the exposure time per image is 200 ms. The channel length is ~150 μm. $z(t)$ represents the PL quenched areas. The time-dependent $z(t)$ is displayed in Supplementary Figure 1. The scale bar represents 100 μm. The color bar is a gray value with arbitrary units, indicating the PL intensity. **g, h** Electrical current $I$ and $1/I^2$, respectively, monitored as a function of time during the measurement of experiment **a**

small initial transient due to ion drift and interfacial capacitive contribution[17,18], details are shown in Supplementary Figure 2. A dark front becomes evident in the left side of the channel, which advances toward the right of the channel during the measurement time. Slowly with time, the current decreases to stabilize at about ~3 µA after a measurement of 40 s with the front covering ¾ of the channel width. After this time, the advance of the dark front slows down and after 50 s, the PL is almost completely quenched with a few inhomogeneities (Supplementary Movie 1). We should remark that the observation of a sharp moving front has been confirmed in independent devices that all show this general behavior, especially at applied electrical fields similar to those present in working photovoltaic devices. However, the point at which the dark front slows its advance and current saturates varied for different perovskite material systems, which shows the overall sensitivity to defect densities (Supplementary Figure 3). There is a clear relationship between the applied electrical field, the velocity in the advance of the dark front, and the measured current. Different perovskite samples, including mixed halide and MQW perovskites have been analyzed using different conditions to prove its validity and are shown in Supplementary Table 2 and Supplementary Figure 3 and 4.

A variety of effects can be invoked to explain a decrease or increase of PL[1] and include mobile charges[19], mobile ions[11,20], defects, and passivation[14,21], solid-state reactions such as the interaction with iodide-related defects and iodine vapor[22], etc. However, the challenge is to establish the mechanism that causes the existence of a sharp front that moves at velocity v. This effect cannot be provoked by removal of electronic carriers swept by the electrical field that would produce a homogeneous decrease of PL across the film. The low velocity of the sharp front indicates that we must consider the displacement of ions. Diffusion of ionic species ascribed to the ion concentration gradient, makes one assume a gradually decaying spatial distribution of PL[23] rather than an advancing edge toward an opposite electrode.

In the following, we develop a quantitative model indicated in Fig. 2 based on the idea that the current across the film (except for a short initial transient effect) is predominantly electronic. The reason for this is that the electronic mobility in hybrid perovskites is about ten orders of magnitude larger than the ionic mobility. However, the applied electric field has the effect that ionic defects will move and change the local charge distribution in the bulk of the sample, decreasing and sometimes increasing the overall current density. In addition, the moving defects promote local solid-state reactions that enhance non-radiative recombination producing a dark area.

**Physical model for the advancing front velocity.** In general, clarifying the dominant moving species in a mixed ionic–electronic conductor is quite challenging. Starting from the seminal work of Azpiroz et al.[24], it has been recognized that iodine vacancies readily diffuse in the perovskite, much faster than MA and Pb vacancies. Several previous works have analyzed the perovskite films with lateral symmetric contacts and studied the quenching of PL when a large bias is applied. These studies concluded that iodide vacancies are the dominant ionic species that moves under the influence of an electrical field, in combination with an electronic current that flows through the sample. Huang et al.[16] observed a moving thread in MAPbI$_3$. Ho-Bailie et al. reported the growing regions of quenched PL dependence on the size of electrical field and humidity[10,11]. In all these cases, the sample becomes darkened at the anode as in our experiments. A recent work by Senocrate et al.[25] provided an interpretation of moving ionic species based on a number of techniques. It was concluded in all cases that iodide vacancies are the main moving

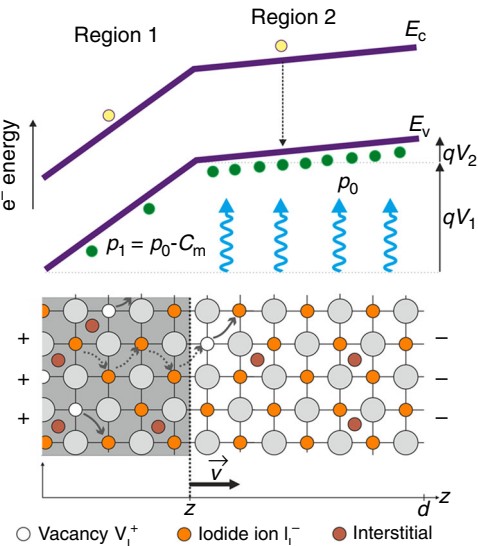

**Fig. 2** Scheme of a physical model. Optoelectronic effects of iodine vacancies drifting in a perovskite layer with symmetric contacts, under the applied field caused by positive bias at the left side. In this model of a p-doped material, the dominant electrical current flowing across the symmetrical electrodes, is an electronic current. The unshaded region 2 shows the original situation of majority carrier holes of density $p_0$ (due to the intrinsic doping by iodine interstitials $I_i^-$) that cause PL under photogeneration of electrons by light soaking. The applied voltage induces an ion movement: iodine vacancies $V_I^+$ drift in the electrical field toward the right. In the shaded region 1, they fill the space up to a density $C_m$ compensating the p-doping and reducing majority carrier density to $p_1 = p_0 - C_m$. With applied voltage V, the conduction band $E_c$ and valence band $E_v$ bend by an amount $qV = qV_1 + qV_2$, where q is elementary charge. The ionic current is negligible compared with electronic current (due to the enormous difference of ionic and electronic carrier mobilities) but ion displacement modifies the local extent of doping and influences the majority carrier density and thus the total current changes (decreases) with time. Due to the drift of ions from the contact, the shaded region grows with time, so that point z advances with time at a velocity v

species. However, the main ionic conductivity component occurs by hopping conductivity of iodine vacancies and not by iodine interstitials. Transient luminescent phenomena due to ionic rearrangement were reported by Bolink et al.[26] and Ginger et al.[27], where the effect of electronic carriers is studied by modification of the injection properties of the contacts. It was shown that in order to have PL quenching in the lateral configuration as a function of the applied bias, not only ion migration is needed but also the presence of electronic current injected from the contacts.

In order to provide a very direct evidence of the p-type character of the herein-used perovskite films, we determined the Seebeck coefficient (Supplementary Figure 5). The sign of the Seebeck coefficient directly points out the p-type majority charge carriers prevalent in a semiconductor and was measured to be 3.4 ± 0.5 mV K$^{-1}$. This value is consistent with previous work by Ye et al.[28] In addition, after biasing 50 V for 180 s, the Seebeck coefficient increases with a factor of 2, implying a change of doping by an external field. Therefore, electronic conductivity is mainly attributed to electronic holes.

Based on the ionic–electronic properties of MAPbI$_3$, we formulate a dynamic model that is outlined in Fig. 2. Here, we propose that a high concentration of $V_I^+$ vacancies is present at the interface with the contacts as a result of lead being undercoordinated by iodine atoms. These types of

undercoordinated species have previously been detected and identified by absorbance measurements[29]. As mentioned previously under a constant applied bias, iodine vacancies $V_I^+$ drift in the electrical field toward the opposite electrode. The additional vacancies correspond to iodine ions that pile up at the contact surface, in a characteristic accumulation region that has been observed in many cases by IS[30] and photovoltage decays[31]. When the $V_I^+$ vacancies move to the right, they compensate the negative charge from iodine interstitials reducing the hole density. Vacancies can fill the space up to a maximum concentration value $C_m$, reducing hole density to $p_1 = p_0 - C_m$ (assuming $p_0 > C_m$). The reduction of doping and the creation of non-radiative recombination sites induces a dark region 1 of size $z(t)$ that increases with time. Meanwhile, we assume that region 2 ($z \leq x \leq d$) remains largely undisturbed with approximately the initial hole density $p_0$. In the case that photogenerated carriers exceed the majority carrier density, $p_0$ indicates the density in the bright region and $p_1$ refers to the lower carrier density in the dark region due to enhanced recombination.

The electrical current in each region $j_i$ ($i = 1, 2$) is driven by the respective electrical field $E_i$ and since the mobility $\mu_p \gg \mu_C$ the ionic current component can be neglected. We have

$$j_1 = q p_1 \mu_p E_1 = j_2 = q p_0 \mu_p E_2 \qquad (1)$$

Hence, the division of voltage between the two regions in $V = V_1 + V_2$ is governed by the equation

$$V_2 = \frac{p_1}{p_0} \frac{d-z}{z} V_1 \qquad (2)$$

and we obtain

$$V_1 = \left[ 1 + \frac{p_1}{p_0} \left( \frac{d}{z} - 1 \right) \right]^{-1} V \qquad (3)$$

The ionic drift current in region 1 is

$$j_C = q C_m \mu_C \frac{V_1}{z} \qquad (4)$$

The ionic flux causes the increase of region 1 as follows:

$$j_C = q C_m \frac{dz}{dt} \qquad (5)$$

Combining the previous expressions, we obtain the equation for the variation of $z$

$$v = \frac{dz}{dt} = z^{-1} \left[ 1 + \frac{p_1}{p_0} \left( \frac{d}{z} - 1 \right) \right]^{-1} \mu_C V \qquad (6)$$

By integration, we get the dependence on time of the velocity of advance of the front

$$v(t) = \left( 1 + \frac{2\gamma}{d} v_0 t \right)^{-1/2} v_0 \qquad (7)$$

where the initial velocity is

$$v_0 = \frac{p_0}{p_1} \frac{\mu_C V}{d} \qquad (8)$$

and we have introduced the quantity

$$\gamma = \frac{p_0}{p_1} - 1 \qquad (9)$$

The electrical current depends on time as

$$j = \left( 1 + \frac{2\gamma}{d} v_0 t \right)^{-1/2} j_0 \qquad (10)$$

where the initial current is

$$j_0 = q p_0 \mu_p \frac{V}{d} \qquad (11)$$

Accordingly, the following representation of $j(t)$

$$\frac{1}{j^2} = \frac{1}{j_0^2} + at \qquad (12)$$

forms a straight line up to the point where current saturates. The slope in Equation (12) is given by

$$a = \frac{2\gamma v_0}{d j_0^2} \qquad (13)$$

The final current at $z = d$ has the value

$$j = \frac{1}{1+\gamma} j_0 = q p_1 \mu_p \frac{V}{d} \qquad (14)$$

It is smaller than the initial current, since the initial amount of $p$-doping has decreased by the spread of vacancies. That the doping has changed is also reflected in the increased Seebeck coefficient after biasing, which increases from $3.4 \pm 0.5$ to $6.2 \pm 0.8$ mV K$^{-1}$ (Supplementary Figure 5).

Let us consider alternative scenarios for the modification of defect distribution in the biased film. Instead of a $p$-doped system assumed above, we take a system that is $n$-doped with intrinsic density $n_0$ and undergoes the same experiment. The model is very similar to Fig. 3, but in this case, the moving iodine vacancies will reduce the doping of region 1 and this zone will get the larger electrical field. The results are given by the same previous equations but now

$$\gamma = \frac{n_0}{n_1} - 1 \qquad (15)$$

Thus, the current increases with time, in contrast to the case above, as now the overall doping is increased. Finally, we consider the case in which the drifting vacancies change the type of doping so that $n_1 = C_m - p_0$. Then region 1 and 2 conduct by electrons and holes, respectively, and the layer operates in a diode mode by double electronic carrier injection from electrodes. The currents meet at $z$ establishing a strong recombination zone at the edge, as in a light-emitting electrochemical cell (LEC), where the redistribution of ionic charges forms the recombination diode structure[32]. Indeed, some perovskite systems show an increased photoluminescence right at the boundary between regions 1 and 2 (Supplementary Figure 6). This observation could indicate the presence of electroluminescence, i.e., the direct recombination of holes and electrons occurs at that place and electron conduction occurs as well, however, it can also be explained by a decrease of non-radiative recombination as commented later. In the herein-

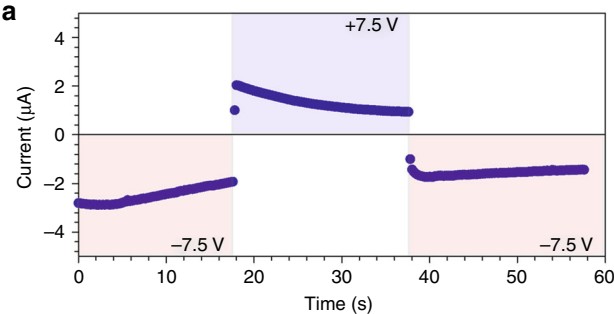

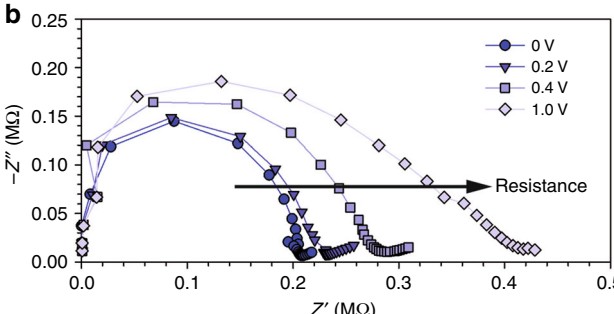

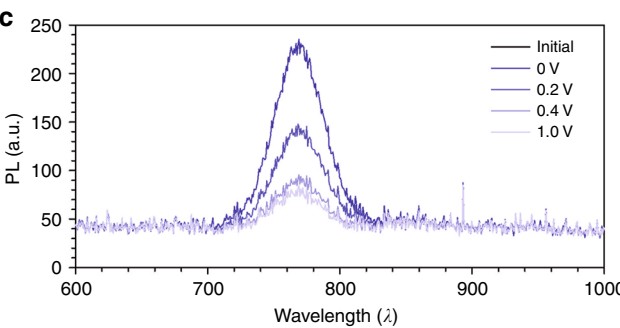

**Fig. 3** Physical measurements of the biased perovskite films. **a** Current decay curves as a function of the polarity of the applied voltage on a CH$_3$NH$_3$PbI$_{3-x}$Cl$_x$ perovskite film with ± 7.5 V across an ~150 μm channel. A change in polarity maintains the current with a different sign, indicating that ionic transport is modifying the bulk properties of the perovskite. **b** Complex impedance plot of an interdigitated electrode measured in the dark as a function of the applied DC bias. **c** PL decay in intensity of the device measured in **b**

presented model, it was completely sufficient to choose a single carrier model of Fig. 2 to describe the experiments.

As it is observed in Fig. 1g that the electrical current decreases with time, we are in the situation of majority hole carriers, where bias-induced excess V$_I^+$ in region 1 reduces the carrier density. Plotting the current as $j^{-2}(t)$ in Fig. 1f, an excellent agreement is obtained with the model, as the figure displays a straight line in the first half of the length $d = 150$ μm. We obtain the results $j_0 = 12$ μA, $a = 4.6 \times 10^{-3}$ μA$^{-2}$ s$^{-1}$, $v_0 = 10.5$ μm s$^{-1}$, and γ = 3.3. By Equation (9), it follows that

$$p_1 = \frac{1}{1+\gamma}p_0 = 0.23p_0 \qquad (16)$$

This result indicates that defects traveling to the bulk layer create a 30% decrease of doping density, as it is also evident comparing the initial and final current density. Using Equation (8), we obtain the mobility of iodine vacancies $\mu_C = 9.1 \times 10^{-7}$ cm$^2$ V$^{-1}$ s$^{-1}$ that corresponds to a diffusion coefficient

$D_{V_I} = 2.4 \times 10^{-8}$ cm$^2$ s$^{-1}$. The current analysis has been applied to different samples, as shown in Supplementary Table 2 and Supplementary Figure 8, and calculated $D_{V_I}$ ranges between $5 \times 10^{-8}$ and $6 \times 10^{-9}$ cm$^2$ s$^{-1}$ values are in good agreement with the value $2 \times 10^{-9}$ cm$^2$ s$^{-1}$ of Senocrate et al.[25]. It is important to note that in some samples, there is a deviation from the modeled curve after half of the electrode is dark and the measured current either slows down the reduction or even increases. The reason is that second region 2 cannot be assumed to remain undisturbed with $p_0$ at this point.

**Alternative interpretations**. In previous work, the interpretation of the front advance in MAPbI$_3$ was attributed to direct migration of ionic defects[11]. Using the velocity of the front, the calculation provides a value $D_{V_I}$ of the order $10^{-11}$ cm$^2$ s$^{-1}$. This result disregards the local charge compensation that influences electronic current and consequently it is rather low. In contrast to this, applying our model based on local saturation effect, we obtain results in agreement with other techniques where $D_{V_I}$ ranges between $5 \times 10^{-8}$ and $6 \times 10^{-9}$ cm$^2$ s$^{-1}$, as pointed out before.

Another approach to explain a decaying current across a mixed electronic–ionic conducting film is the method of stoichiometric (galvanostatic) polarization, which assumes that the initial current is mixed electronic and ionic, while the final current is only electronic. However, these experiments are usually performed[25,33,34] at lower current density. It has been shown that the existence of a moving front that creates a dark region requires a large applied electrical field, higher than a threshold value, that is a function of ambient humidity[11]. It is likely that galvanostatic polarization conditions[25,33,34] should occur below the threshold for the formation of a dark region advancing front. In the other extreme, a larger applied electrical field may damage the MAPbI$_3$ sample[11,35]. It is worthwhile to mention that CsPbI$_2$Br shows enhanced stability to an applied electrical field and larger activation energies for ion hopping[34].

**Evidence on the dominance of electronic current**. There are two experimental evidences that suggest that the measured current in our experiment is due to the change in the electronic conduction properties of the bulk perovskite layer under illumination, rather than to the disappearance of an ionic contribution. On the one hand, when the applied voltage is reversed, the current is maintained just with a change of sign, Fig. 3a. These results clearly indicate that the dominant current is electronic current passing through the film, as opposed to a significant ionic contribution, as reversing the applied voltage would instantaneously increase the initial transient current due to accumulated ions. Different examples of voltage reversal are shown in Supplementary Figures 8, 9, and 10. It should be pointed out also that electronic current in laterally contacted electrodes has been observed to increase with time[27], which is impossible in stoichiometric polarization conditions but quite likely in our model of ion-induced doping modification in the bulk. On the other hand, IS measurements have been carried out with the double function of providing a source of electrical field, by the application of a DC bias, and to probe the electrical effects that take place at different time scales. In this measurement, the high-frequency signal is due to the properties of the perovskite layer[2]. Interestingly, the arc at high frequency increases the resistance as the PL intensity decreases by the application of increasing potential bias, Fig. 3b, c. Overall, these evidences point to a modification of the bulk properties of the perovskite layer as responsible to the modification of the electronic properties.

## Discussion

In this section, we analyze in detail the mechanism whereby the moving ionic defects provoke a total disappearance of PL in a part of the sample. Since the first observations of PL quenching in biased and light-soaked perovskite films, mobile ion-induced enhanced non-radiative recombination[10] has been suggested as the main cause of film darkening in regions that grow with time. So far, it has not been explained mechanistically how the advancing front of the iodide vacancies causes the supresion of PL. It is natural to assume that the ionic flux that advances from the positive electrode consisting of anion vacancies is starting a local solid-state ionic reaction that creates abundant non-radiative recombination centers. While the opposite ionic reaction occurs often at the edges, removing the non-radiative local species, and enhancing, rather than decreasing, PL.

First of all, we must analyze the type of defects that can be present in our samples. In this respect, Seebeck effect clearly points to $p$-type doping in the initial state. This type of doping can be due to either an excess of interstitials negatively charged ions ($I^-$), vacants of positively charged native ions ($MA^+$, $Pb^{2+}$), or ions placed in the lattice arising from redox reactions that somehow are able to stabilize the perovskite structure (i.e., Pb (0)). From X-ray photoelectron spectroscopy (XPS) measurements, metallic lead Pb(0) is frequently observed, as a degradation product. Therefore, Pb(0) should be stable in the film[36]. These species would include $I_i$, $V_{MA}$, $V_{Pb}$, and lattice Pb(0).

The origin of these defects is not central in our work but one can attribute them to the generation of Schottky or Frenkel defects. Our observations are compatible with the presence of Frenkel defects involving displacement of an atom from the lattice position to an interstitial position and generation of a vacancy. Other works have endorsed the conclusions of these observations[37,38] and the presence of iodine interstitials have recently been measured by X-ray and neutron scattering[39]. Schottky defects indeed have been shown to have reasonably small formation energy[40]. However, very recently, Kelvin probe microscopy experiments using a similar lateral device geometry as in this work, show the major prevalence of Frenkel defects and their long-range motion in an electric field[27,41].

Next, we must consider the species that can be present in our films and could lead to PL quenching. There are different species that have been reported to quench the PL in the perovskite film forming effective recombination centers[38]. Iodide interstitials have been recently highlighted as one predominant non-radiative recombination center. A second possibility is the presence of Pb vacancies. However, due to the large size of the ion, this quenching ability of Pb would be expected to be observed as background for all our experiments, as it is not expected that the electrical field would lead to migration of the ion. Alternatively, formation of interstitial Pb(0) has also been suggested by Birkhold et al. as the dominant recombination center[27]. Furthermore, the oxidation state of the halogen vacancy has a dominant effect on the extent of recombination at a defect site[42].

Finally, we can provide an interpretation of migrating species and its relationship with PL quenching, outlined in Fig. 4. We have an initial background doping due to any of the species described above ($I_i$, $V_{MA}$, $V_{Pb}$, and lattice Pb(0)). We can assume that all of them will be present in our films. From these species, only $I_i$ will be responsible for PL quenching, as suggested by density functional theory (DFT) calculations. The initial PL intensities of the samples are high, but in many cases, the PL increases with the time under illumination (and no applied electrical field). Then, we cannot rule out the presence of charged $I_i$ as one of the initial dopants that could be neutralized by the reverse of Frenkel-type defect chemistry, this is the combination of vacancy and iodine interstice to eliminate the quenching site. As vacancies advance toward the negative electrode, they compensate the charge of the background doping induced by $I_i$, $V_{MA}$, $V_{Pb}$, or Pb(0) reducing the conductivity. In addition, the excess iodine vacancies (compensated by electrons) can lead to redox reactions as those described by Birkhold et al., in which interstitial $Pb^{2+}$ is reduced to Pb(0)[27]. Both $I_i$ and interstitial Pb(0) will be the recombination centers that lead to the dark area in the PL. Overall, the positive electrode will be enriched of iodine atoms, as supported by XPS measurements carried out previously by our group[43].

It can be observed in different images that there is a bright edge close to the contacts, indicating that the actual species present at the contacts are different to those in the bulk, with less concentration of PL quenching centers close to the contacts. Here, we propose that at the bright edge, there is a large concentration of $V_I$ susceptible to be transported under an applied electrical field

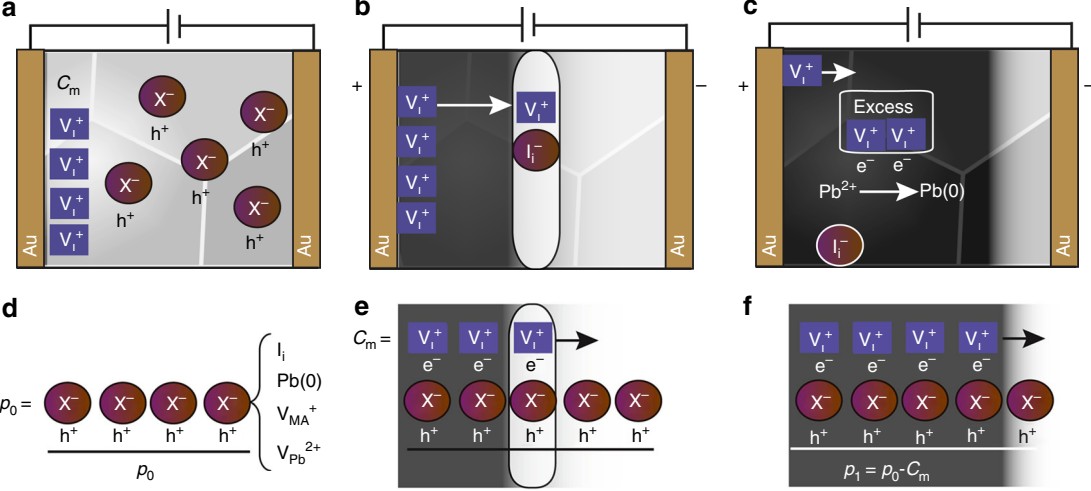

**Fig. 4** Interpretation of the dominant ionic effects. Diagrams with a proposal of the processes taking place during the application of an electrical field in the lateral electrode configuration. **a** and **d** represent the initial state with a background of $p$-dopants ($p_0$) and iodine vacancies present at the interface with the electrode. **b**, **e** When the electrical field is applied, vacancies migrate toward the negative electrode, in this transit, they partially compensate the charge of the background doping, including some non-radiative recombination defects ($I_i$). **c**, **f** Excess vacancies carry excess of electrons that can lead to generation of other PL quenching defects such as Pb(0)

as dominant moving species, as proposed by Senocrate et al.[25] and low concentration of $I_i$ as quenching species. As can be observed in Supplementary Movie 1 (from 8 s onward), at the edge of the advancing front, there is an increase in PL which could be the result of neutralization of $I_i$ recombination centers by $V_I$, just the opposite to Frenkel defect generation.

It is important to highlight that the direction of the advancing front will be highly dependent on the actual doping species present in the films. Our results clearly show an advance of the dark front from the positive to the negative electrodes in agreement with different authors[11,44]. Alternatively, there are other authors who report that the dark front advances from negative to positive[12,27,45]. If the $V_I$ are initially abundant, the negative electrode will be enriched with vacancies and when these are confined at the electrode, they can lead to formation of $PbI_2$, as described by Huang et al.[16] in agreement with a Pb/I ratio lower than three, also supported by XPS[43]. Indeed, it has been reported that the doping nature of the $MAPbI_3$ will strongly depend on the processing conditions to prepare the films and imbalance of the reagents with excess of $MA^+$ or $PbI_2$ can lead to n-doping or p-doping, respectively[46].

In conclusion, an electrical bias in a perovskite layer can switch on and off the photoluminescence in a timescale of seconds. We explain the advance of the dark area with a sharp front in terms of the drift of iodine vacancies that fill the space up to a critical density turning the material more intrinsic-like and enhancing non-radiative recombination. An interplay between interstitial and vacancy defects determines the dominant electronic density and subsequently allows a control of electro-optical properties.

The model fits perfectly with the experimental data describing the initial dynamics of the drift of vacancies in an electric field and directly allows us to determine the mobility of iodine vacancies. This method provides a direct and visual way to track the migration of vacancies and obtain a value for the defect density and is applicable to a broad range of organometal halide perovskite systems as well as Ruddlesden–Popper halide layered perovskites. The implications are significant as it shows that vacancies (i.e., ions) redistribute in the presence of an electric field as it is prevalent in a solar cell device and change the respective electron/hole densities affecting the overall electronic current. It will greatly help to understand how to compensate defect and migration processes in order to eliminate hysteresis effects and device degradation.

## Methods

**Device fabrication**. $CH_3NH_3PbI_{3-x}Cl_x$ precursor solution was prepared by dissolving 0.88 M lead chloride ($PbCl_2$) (Sigma-Aldrich) and 2.64 M $CH_3NH_3I$ (MAI) (Tokyo Chemical Industry company) in anhydrous N,N-dimethylformamide (DMF) (99.8%, Sigma-Aldrich). The precursor solutions were filtered through a 0.2 μm polytetrafluoroethylene (PTFE) filter (Carl Roth GmbH + Co. KG). Glass substrates were washed with acetone and isopropanol for 10 min each. Then these glass substrates were treated with ozone for around 10 min. In a nitrogen gas-filled glovebox, this precursor solution was spin-coated on the glass substrates at 3000 rpm for 60 s. Following that, the as-spun films were annealed at 100 °C for 80 min in the glovebox. The characterization of the perovskite film is shown in Supplementary Figures 11 and 12. The performance of the photovoltaic device using this perovskite film is shown in Supplementary Figure 13. The perovskite film on glasses was transferred into an evaporation chamber with pressure of ~$3 \times 10^{-6}$ mbar, and ~70 nm thickness of Au was deposited by thermal evaporation through a shadow mask. This interdigitating shadow mask defined the electrode geometry: the electrode distance was 200 μm and a ratio between channel width W and length L was of 500. External conducting wires were connected to the device using an Ultrasonic Soldering System (USS-9200, MBR electronics GmbH). In the end, to protect the film from oxygen and water, 40 mg mL$^{-1}$ poly(methyl methacrylate) (PMMA) solution dissolved in butyl acetate (anhydrous, 99%, Sigma-Aldrich) was spin-coated on the film at a speed of 2000 rpm for 60 s in the glovebox.

**Photoluminescence imaging microscopy**. This measurement was conducted on a home-build PL microscope, as shown in Supplementary Figure 7. Based on a commercial microscopy (Microscope Axio Imager.A2m, Zeiss), the sample was allocated

in the focal plane of an objective lens (10 × /0.25 HD, Zeiss), and the sample position was adjusted by a motorized scanning stage (EK 75*50, Märzhäuser Wetzlar GmbH & Co. KG). The sample was illuminated by an internal LED illuminator using a dichroic mirror and filter combination (HC 440 SP, AHF analysentechnik AG) with the excited wavelength of around 440 nm. The excitation light intensity can be controlled and was set to ~35 mW cm$^{-1}$. The PL signal was filtered (HC-BS 484, AHF analysentechnik AG) to suppress residual excited light and directed onto a CCD camera (Pco. Pixelfly, PCO AG) with the exposure time of 200 ms. A constant voltage of 4–10 V was applied between the Au electrodes (236 Source Measure Unit, Keithley Company), and the current was monitored and recorded by a LabVIEW program. During the measurement, the excitation illumination was kept on continuously. Under the above illumination and bias condition, the ratio between light current and dark current was measured as $I_{light}/I_{dark}$ = ca. $10^2$. Additional measurements coupled PL imaging with impedance spectroscopy measurements using an Autolab PGSTAT 204 with an impedance analyzer module. Measurements were carried out using DC bias, as described in the text with a 20 mV (RMS) AC voltage perturbation with frequencies ranging from 20 mHz$^{-1}$ to 1 MHz.

**Seebeck coefficient measurements**. The Seebeck coefficient was measured on perovskite films, which were processed in exactly the same way as above. As shown in Supplementary Figure 5(b), lateral single-channel gold electrodes with a spacing of 1 mm were evaporated maintaining the same sample geometry. The sample was placed on a differential temperature stage with a hot and a cold side enclosed in a small vacuum chamber at 0.1mbar. The temperature on one side was changed between 50 and 75 °C while keeping 24 °C on the other side and an equilibration time of 20 min was allowed. The thermal voltage was measured with an Autolab PGSTAT 204 averaging over 300 single measurements.

## Data availability

The experimental data that support the findings of this study are available from the corresponding author upon reasonable request.

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

## Acknowledgements

We acknowledge funding from MINECO of Spain under Project MAT2016-76892-C3-1-R. A.G. would like to thank the Spanish Ministerio de Economía y Competitividad for a Ramón y Cajal Fellowship (RYC-2014-16809). C.L and S.H. gratefully acknowledge the financial support by the Bavarian State Ministry of Science, Research, and the Arts for the Collaborative Research Network "Solar Technologies go Hybrid" as well as the German Research Foundation (DFG). Part of this research was undertaken on the SAXS/WAXS beamline at the Australian Synchrotron, Victoria, Australia. Furthermore, we are very thankful to Shannon Yee, Bernd Kopera, and Marcus Retsch for their dedicated support with the Seebeck coefficient measurements. C.L. thanks Xudong Cao and Yu Zhong for the GIWAXS, XPS, and XRD characterization, Dr. Thomas Unger, Dr. Fabian Panzer, and Dr. Julian Kahle for optoelectronic characterization and discussion.

## Author contributions

C.L. and A.G. conducted the experiments and coordinated the experiments. J.B. proposed the model and prepared the first draft of the manuscript. S.H. and J.B. supervised the work. All authors co-wrote, discussed, and commented on the paper.

## Additional information

**Competing interests:** The authors declare no competing interests.

