## [Peer Review File · Nature Communications]

Reviewers' comments:

Reviewer #1 (Remarks to the Author):

Using optical microscopy Li et al observed bias electric field induced PL on/off phenomenon in halide perovskite planar film with lateral symmetric electrodes. They confirmed that a dark region advances from the positive electrode at a slow velocity of order of $1\mu\text{m s}^{-1}$. They intended to address the sharp front moving in terms of the drift of ionic vacancies that drastically reduce the PL efficiency. They proposed a dynamic transport model and showed that the square reciprocal of the electrical current is linear with time.

Basically, this topic can be interesting because the observed phenomena are not yet fully or quantitatively understood to date. However, relative to the previous publications, the current manuscript does not provide significantly advance; and there is significant problem in this manuscript. To be considered for publication in Nature Communications, I think authors must carefully address the following concerns and significantly improve the manuscript:

1. Authors ignored the similar phenomena previously observed by other groups, for example, Deng et al. Electric field induced reversible and irreversible photoluminescence responses in methylammonium lead iodide perovskite, *JMC-C* 4(2016) 9060; Yuan et al. Electric-Field-Driven Reversible Conversion Between Methylammonium Lead Triiodide Perovskites and Lead Iodide at Elevated Temperatures, *Advanced Energy Materials* 6 (2), 2015.
2. "ionic vacancies drastically reduce the PL efficiency"? Authors need to make deep insight for the mechanism for ion induced PL quenching, although some previous publications, for example, Wen et al. Mobile charge-induced fluorescence intermittency in methylammonium lead bromide perovskite, *Nano letters* 15 (2015), 4644
3. In experiment, the PL image is essential data. It is important to use the consistent scale bar for PL intensity, in addition the size. Authors should clearly interpret the relation of PL intensity variation—spatial—time.
4. Illumination effect cannot be ignored, many papers already mention the mobile ion activation by the illumination, in addition to the electric field. The detailed illumination condition should be presented, such as during, pulse or continuous, .
5. Some statement lack evidence or justification, for example, "These findings were described in terms of a simple mechanism common to both fully inorganic and organic semiconductors (6, 7) in which photogenerated electrons and holes drift to opposite sides of the device, reducing the bulk recombination rate and hence PL intensity." and "However, diffusion of the vacancies would cause a gradually decaying spatial distribution of PL rather than an advancing edge."
6. Can authors provide a few simulations according the modelling? and explain the sharp edge shift.

Reviewer #2 (Remarks to the Author):

While the selected type of experiment described here is interesting and able to reveal much information on the materials, it has to be pointed out that this work suffers from a variety of shortcomings:

- 1) The first point refers to novelty and originality. The authors essentially refer to experiments (with a slightly different composition) and qualitatively similar results that they already previously published in [Li C., Guerrero A., Zhong Y., Gräser A., Luna C. A. M., Köhler J., Bisquert J., Hildner R. and Huettner S., Real-Time Observation of Iodide Ion Migration in Methylammonium Lead Halide Perovskites. *Small* 13, 1–10 (2017)]. Although they couple these with a dynamic transport model,

added value of this paper is not sufficient to justify publication in Nature Communication.

2) Notwithstanding the fact that the authors focus on the effect of anion vacancies on the PL response of MAPbI₃, they strongly suggest that iodine interstitials are present in large number in the MAPbI₃ structure (generated by a Frenkel defect reaction). This claim is not supported by any evidence, especially considering that normally perovskites are dominated by Schottky defect reactions (metal and anion vacancies). The authors feel supported by recent measurements (Ref. 17 - Minns, J. L., Zajdel, P., Chernyshov, D., van Beek, W. & Green, M. A., Nat. Commun. 8, 15152 (2017)). However, in that report such interstitials are neutral (and therefore of much different size than negatively charged iodide ions) and form molecular aggregates. Such defects would not contribute to p-type conductivity, as claimed by the authors, but rather would represent a loss in electron holes, opposite to their claims and observations. On the same lines, the authors neglect the significant amount of literature calculating low formation energies for Schottky defects (for example, [1] Eames, C. et al., Nat. Commun. 6, 7497 (2015) -- [2] Haruyama, J., Sodeyama, K., Han, L. & Tateyama, Y., J. Am. Chem. Soc. 137, 10048–10051 (2015) –[3] Walsh, A., Scanlon, D. O., Chen, S., Gong, X. G. & Wei, S.-H., Angew. Chem. Int. Ed. 54, 1791–1794 (2015)).

3) The authors model the current behaviour of their samples by assuming the ionic current to be negligible. Simply by looking at Fig. 1c, this is evidently an unwarranted claim. Instead, it appears very clear how the ionic current (which is expected to decrease as a function of time due to progressive ion blocking) makes up the largest part of the total current during the initial biasing. Indeed, the value starts at 7 μ A (ionic and electronic), while at 120 seconds it is decreased to 2 μ A (only electronic). Such a claim could only be valid at long times, where the ionic current disappears, but cannot describe the development of the bias-induced polarization.

4) The authors refer to ion transport in halide perovskite, but it appears they do not consider the full implications of such transport on these materials. Specifically, the authors only discuss ion accumulation at the interfaces, rather than considering that such polarization must affect the entire bulk of the material via effectively neutral stoichiometric variation, which is a basic phenomenon in solid state electrochemistry (example references are: [1] Wagner, C., Ber. Bunsenges. Phys. Chem. 60, 4–7 (1956). -- [2] Hebb, M. H., J. Chem. Phys. 20, (1952). -- [3] Yokota, I., J. Phys. Soc. Japan 16, 2213–2223 (1961).).

In summary, it appears that the results presented here suffer from several major scientific issues and apart from that do not provide sufficient advancement on the present knowledge or enough novelty. I therefore cannot recommend publication in Nature Communications.

Reviewer #3 (Remarks to the Author):

In this manuscript, Cheng Li et al. report on the electrical switching of photoluminescence in lead halide perovskite and explain the underlying mechanism based on a simple dynamic transport model. In brief, the authors attribute the existence of the sharp front and movement of the sharp front of the luminescence to the drift of the ionic vacancies, which would reduce the doping and create the nonradiative recombination center leading to the reduced luminescence. They further extracted the ion diffusion coefficient based on the simple model they developed. Overall, the topic is interesting and important; however, the model is oversimplified by neglecting too much factors such as built-in electric field and experimental results cannot fully support their conclusions. Therefore, the referee cannot recommend its publication before the authors fully address the following questions.

1 The description is too simple and not clear at all for Figure 1. The authors are suggested to indicate the sharp front in Figure 1a and explain why the luminescence at 30s and 55s is so weak. If possible, it would be better to provide a video to show the movement of the sharp front and the change of the luminescence intensity.

- 2 The authors claimed that the recovery of ionic vacancies occurs at a much slower rate after removing the bias voltage or basing in the contrary direction. How slow is this process? How does the current evolve after removing the bias? Can the authors provide more information on this?
- 3 The model the authors developed is based on the assumption that MAPbI₃ is p-doped at room temperature. Nevertheless, the authors did not provide any evidence. As a matter of fact, previous transport studies have demonstrated that MAPbI₃ at room temperature is n-doped (see Nature Communications, 7, 11330, 2016; Nature Communications, 6, 7383, 2015). Can the authors provide transport measurement to prove that the films they used are p-doped?
- 4 The authors attributed the decrease of the current and luminescence intensity to the drift of the ionic vacancies, which would reduce the doping and create the nonradiative recombination center leading to the reduced luminescence. Nevertheless, they did not provide any information on the concentration of the iodine vacancies and iodine interstitials. The decrease of the measured current is on the order of μA . Can the ionic drift reduce the hole concentration on such large degree?
- 5 In the model the authors developed, they assume the electric field is uniform cross the region 1 and region 2 and neglect the built-in field due to the uneven distributed ions. Can the author justify this? The built-in field due to the ions is expected to be quite large if the concentration of ions is such large that can reduce the doping concentration so much.
- 6 Can the authors provide PL spectra simultaneously with PL images shown in Figure 1a? The evolution of the peaks of PL spectra might be able to show useful information to support the authors assumption.
- 7 The authors claimed that 'When iodine vacancies move to the right they compensate the negative charge from iodine interstitials reducing the hole density as shown in Figure 1e.' How can iodine vacancies compensate the negative charge from iodine interstitials? Do the interstitials capture the iodine vacancies? Can the authors illustrate more on this using schematic illustrations?
- 8 The authors stated that the drift of iodine vacancies can create the nonradiative recombination sites. Can the authors explain how to achieve this?
- 9 In equation (15), p_0 should be n_0 . In Figure 2, the change of the energy band would not expect to change so suddenly due to the existence of the build-in field.

Reply to reviewers

General statements

We would like to start with some general statements in order to avoid repetition, thereafter we provide a point-point response to the reviewers. We are very thankful to the referees for their detailed reading and thoughtful critiques that have been very important for us to considerably improve this paper. We took the comments very seriously and have worked intensively in the last months in order to develop a robust set of new data and measurement techniques, a more consistent picture and consequently more clarity of interpretation. In order to improve the presentation and clarify some important remarks by the referees, we have now included significant new points of evidence and discussion and we think these will be rather convincing.

In this paper we wish to explain quantitatively the observation of a moving dark area when a laterally contacted sample is biased under light soaking. We develop a quantitative model based on the idea that the current across the film is predominantly electronic (except for a short initial transient effect. The dominant reason for this is that the electronic mobility in hybrid perovskites is about 10 orders of magnitude larger than ionic mobility. The applied electric field has the effect that ionic defects will move and change the local charge distribution in the bulk of the sample, decreasing and sometimes increasing the overall current density. In addition the moving defects promote local solid state reactions that enhance nonradiative recombination producing a dark area.

The new information included in the revised manuscript with respect to the first version is the following:

1. **A broad new set of data of measurements of new films.** The new extended investigation with a broad set of compositions has shown that the behaviour of slow advancing front is a characteristic behaviour. The model suggested in this paper has been applied with good results. A very broad set of results of 6 different types of perovskite samples is reported in the Table 1 of SI, Fig S1, S3 and S4 present the shape of dark regions, and we have included 3 videos in SI showing the advance of the dark region. In addition it must also be recognized that we found a number of cases where the advancing front was not observed, and other responses such as globular or filament growth of dark regions *via* preferential ionic pathways. Further investigation was restricted to the samples with linear advancing front. A new measurement of the clear advancing front is shown in the new Fig. 1 of the main text, as follows:

Fig. 1

Further experiments are shown in Fig. S3 and S4

2. Measurement of impedance spectroscopy.

A new measurement has been introduced, it is impedance spectroscopy at different bias voltage with simultaneous measurement of PL.

This measurement clearly shows the growth of a more resistive region as the PL decreases. The high frequency signal is due to the properties of the perovskite layer. Interestingly the resistance in the arc at high frequency increases as the PL intensity decreases by the application of increasing potential bias. These evidences point to a modification of the bulk properties of the perovskite layer as responsible to the modification of the electronic properties.

3. Measurement of Seebeck coefficient

We performed Seebeck coefficient measurements on perovskite thin films with a lateral electrode geometry as shown in Figure S7. Results clearly show the p-type behaviour of the herein prepared samples. This is an important information in order to recognize the changes of carrier transport induced by the ionic drift. What is more, we

also measured the Seebeck coefficient after biasing for 180 seconds. The increased Seebeck coefficient implies the change of the doping towards intrinsic semiconductor. In addition, these measurements belong to one of the first Seebeck coefficient measurements which have been carried out on hybrid metal halide perovskites thin film. This will definitely generate a lot of attention within the community.

4. Interpretation of moving defects

The interpretation of moving defects is normally a complex issue but some consensus has been achieved. Starting from the seminal work of Azpiroz *et al.*¹, it has been recognized that iodine vacancies readily diffuse in the perovskite, much faster than MA and Pb vacancies. Several past experiments have analyzed the perovskite films with lateral symmetric contacts and studied the quenching of PL when a large bias is applied. These studies concluded that iodide vacancies are the dominant ionic species that moves under the influence of electrical field, in combination with an electronic current that flows through the sample. Huang and coworkers² observed a moving thread in MAPbI₃. Ho-Bailie and coworkers reported the growing regions of quenched PL dependence on the size of electrical field and humidity.^{3,4} In all these cases the sample becomes darkened at the anode as in our experiments. A recent work by Maier and coworkers⁵ provided an interpretation of moving ionic species based on a number of techniques. It was concluded in all cases that iodide vacancy is the main moving species.

Ginger and coworkers,⁶ (in their work that has been published after our original submission to Nat Comm.) studied the effect of electronic carriers by modification of the injection properties of the contacts. It was shown that in order to have PL quenching in the lateral configuration as a function of the applied bias not only ion migration is needed but also the presence of electronic current injected from the contacts. The electronic flow is a dominant aspect of suppression of PL as we explain in our model. We have assumed that the dark region is primarily induced by the flow of dominant hopping species, which is positively charged iodine vacancies. There is no evidence for ionic flow from the negative to the positive. Normally the negative electrode remains unaffected while the major darkening changes occur from the positive electrode towards the center. However in the work of Ginger and coworkers⁶ the dark region occurs at the

cathode. It is also reported that the current increases with time opposite to our observations. We interpret these observations as a change of the type of electronic conductor. It has been shown that modification in the precursor induces a change from p-type to n-type in the perovskite layer.⁷

5. Summarizing the previous information we have the following results:

- (a) Upon application of a bias voltage the dark region starts at the positive electrode. The negative electrode remains unchanged. The dark region is formed by the drift of positive defects, while there is no evidence of long range drift of negatively charged defects. We assume that the positive defects are iodide vacancies.
- (b) The Seebeck measurement indicates predominant p-doping and change of the Seebeck coefficient after biasing.
- (c) The dark region introduces a lower electronic conductivity as indicated by direct IS measurement, that probes the bulk perovskite film, pointing to p-doping is reduced as the dark front advances.

These three points provide a joint confirmation of the model outlined in the Fig 3 of the paper, namely: The variation of electrical current at intermediate times (*after* the first transient behaviour, and *before* the PL is generally exhausted) is caused by a modification of the electronic flux that goes through the sample between the symmetric electrodes. The transport of positive defects reduces p-doping and decreases the conductivity. At the same time the transport of positive defects induces the generation of more nonradiative recombination centers and quenches PL.

The application of the model allows an excellent fitting of the current in this intermediate regime, as shown in Fig. S11, with results summarized in Table SII. The model allows calculation of ionic mobilities with a great detail and overall provides a good agreement of experiment and theory.

In previous work the interpretation of the front advance in MAPbI₃ was attributed to direct migration of ionic defects.⁴ Using the velocity of the front the calculation provides a value D_{V_i} of the order 10^{-11} cm²s⁻¹. This result obtained by a direct calculation is rather low. In contrast to this, applying our model based on local saturation effect we obtain results in agreement with other techniques where D_{V_i} ranges between 5×10^{-8} - 6×10^{-9} cm²s⁻¹.

6. Ion transient effects and mixed ion-electronic conduction

As pointed out by referee 2, another approach to explain a decaying current across a mixed electronic-ionic conducting film is the method of stoichiometric (galvanostatic) polarization, which assumes that the initial current is mixed electronic and ionic, while the final current is only electronic. However these experiments are usually performed^{5,8,9} at lower current density. It has been shown that the existence of a moving front that

creates a dark region requires a large electrical field, higher than a threshold value, that is a function of ambient humidity.⁴ It is likely that galvanostatic polarization conditions^{5,8,9} should occur below the threshold for the formation of dark region advancing front.

Nevertheless we totally agree with the remark of the referee 2 that there exist strong ionic transient effects and. For example if we look at the main data set in Fig 1 above, we can notice some perovskite/gold interfacial effects. We can clearly see that initially there are boundary layers with excess PL. When the dark zone grows, the left thin PL layer disappears, but the right layer persists unchanged. This last effect is evident in many samples. The video SI V2 shows a very strong luminescence at the boundaries. In these specific regions the defects that cause PL quenching have been severely reduced.

Throughout the experiments we could observe several transients of PL. In some cases the intensity shows a major rise and then decreases. The intensity becomes larger and then decays in some regions. It is characteristic that the PL intensity increases at the boundary between dark and light regions. The following figure clearly shows these inhomogeneous effects. There is intense luminescence at the dark/light boundary, and the overall PL is increasing.

Figure S5: Integrated and normalized PL intensity showing the PL profile within the channel. The quenched PL area is moving from left to right. The PL intensity increases through electroluminescence right at the border between region 1 and region 2.

These effects remain out of the scope of the simple and dominant tendency told in the model of Fig. 2. We believe these effects are mainly caused by the combination of:

6.a Ionic accumulation at interfaces. In addition to the overall flux of vacancies forming a sharp front, the electrical field imposed externally causes ionic accumulation effects at the interfaces and at the front boundary.

6.b. Sweep of electronic charge and variation of charge injection. The change of boundaries and electrical field distribution can affect injection of carriers, or their

removal, and produces changes of PL.

In order to obtain more information we carried out additional experiments to characterize these effects. We studied the inversion of voltage in order to separate transient capacitive of fast ionic rearrangement with the slower dynamics that governs the advance of the dark front. As expected there is a varied phenomenology depending on the applied voltage and inversion time, but some patterns reveal interesting information as shown in the following figures of SI

Fig. S8

Fig. S9

Fig. 3a

First upon inversion of voltage in Fig. S8, it is observed a fast ionic transient peak after which the current starts to grow. This results shows that the advancing front in our experiments cannot be simply described by the suggestion of referee 2 where “the ionic current is expected to decrease as a function of time due to progressive ion blocking”. If we have ion accumulation from previous biasing it is natural to obtain a sudden rise in the current, and this is observed in several instances. However a progressive ion blocking can never increase the current over a long time scale, and this is observed in Fig. S8. In addition Fig. S9 and Fig. 2a show cases where the inversion of voltage does not modify at all the magnitude of the current, which is simply inverted with no transient at all. In this case the switch of voltage simply changes the direction of electronic current while the resistance of the film is the same.

However the issue can become much more complicated. The process can be seen in detail in video V2 in SI. We can see once is changed the direction of the voltage (at around 14 second), there are two process occurring. (1) In the PL dark area, there is a quick redistribution of ions, which takes less than 1 sec in video (less than 4 second in real time). (2) Meanwhile, the opposite electrode begins to form new PL black front and move to opposite direction. Process (1) leads to the increase of the current. This sudden change can be described as ion drift induced capacitance.^{10,11} Process (2) leads to the decrease of the current. The combination or competition of process (1) and (2) results in the strange behavior of the time dependent current.

In conclusion we totally agree with the indication of referee 2 that there are transient ionic current, and we have mentioned it in the main text. However the current density over longer times appears to be dominated by the electronic current going through the film. There is a complex interplay between the strong modification of the film by the advance of vacancies in a sharp front, and further effects of polarization at interfaces.

7. Ionic-induced suppression of PL

Since the first observations of PL quenching in biased and light-soaked perovskite films, mobile ion-induced enhanced nonradiative recombination³ has been suggested as the main cause of film darkening in regions that grow with time.

So far it has not been explained mechanistically how the advancing front of the iodide vacancies causes the suppression of PL. It is natural to assume that the ionic flux

that advances from the positive electrode consisting on anion vacancies is starting a local solid state ionic reaction that creates abundant non-radiative recombination centers. While the opposite ionic reaction, occurs often at the edges, *removing* the non-radiative local species, and enhancing, rather than decreasing, PL.

Given the transient nature of the experiment, the fact that it is based on detection of electrical current that do not *per se* indicate the nature of the moving charged species, and our current experimental tools, we must recognize that it is not possible for us to identify by direct experimental evidence the variation of ionic defects that is changing and causing a drastic suppression of PL. However we will propose a feasible mechanism based on our own results and support it with previous results reported in the literature.

First of all, we must analyze the type of defects that can be present in our samples. In this respect, Seebeck effect clearly points to p-type doping in the initial state. This type of doping can be due to either excess of interstitials negatively charged ions (Γ), vacants of positively charge native ions (MA^+ , Pb^{2+}) or ions placed in the lattice arising from redox reactions that somehow are able to stabilize the perovskite structure (i.e. $Pb(0)$). From XPS measurement, metallic lead $Pb(0)$ is frequently observed, as a production of degradation. Therefore, $Pb(0)$ should be stable in the film.¹² These ions would include I_i , V_{MA} , V_{Pb} and lattice $Pb(0)$.

The origin of these defects is not central in our work but one can attribute them to generation of Schottky or Frenkel defects. The reviewer is absolutely right, that Schottky defects show small formation energies and that they may be prevalent. In fact, in the literature several proposals coexists and there are papers in favour or against each hypothesis. But, very recent studies from 2018 give a more concise picture of the overall presence of Frenkel defects. One example are the experiments by the Ginger group,⁶ who use a very similar device geometry as we do, but include an insulating layer between the contacts and probe the charge density redistribution by Kelvin probe microscopy across the channel. Their experiments clearly support the prevalence of Frenkel defects and a long-range motion thereof. We have refined the manuscript to explain this fact in a better way and we have added various older and very recent references on this area.

Next we must consider the species that can be present in our films and could lead to PL quenching. There are different species that have been reported to quench the PL in the perovskite film.¹³ In the graph below the species with states in the bandgap will lead to non-radiative recombination centers. First of all we note that iodide interstitials have been recently highlighted as one predominant non-radiative recombination center. A second possibility is the presence of Pb vacancies. However, due to the large size of the ion this quenching ability of Pb would be expected to be observed as background for all our experiments as it is not expected that the electrical field would lead to migration of the ion. Alternatively, formation of interstitial $Pb(0)$ has also been suggested by Birkhold and coworkers as the dominant recombination center.⁶

Main defects energy levels, Filippo de Angelis,

Finally, we can provide an interpretation of migrating species and its relationship with PL quenching. We have an initial background doping due to any of the species described above (I_i , V_{MA} , V_{Pb} and lattice $Pb(0)$). We can assume that all of them will be present in our films. However, from these species only I_i will be responsible for PL quenching as suggested by DFT calculations. The initial PL intensities of the samples are high but in many cases the PL increases with the time under illumination (and no applied electrical field). Then, we cannot rule out the presence of charged I_i as one of the initial dopants that could be neutralized by the reverse of Frenkel type defect chemistry, this is the combination of vacancy and iodine interstice to eliminate the quenching site.

As vacancies advance towards the negative electrode they compensate the charge of the background doping induced by I_i , V_{MA} , V_{Pb} or $Pb(0)$ reducing the conductivity. In addition, the excess iodine vacancies (compensated by electrons) can lead to redox reactions as those described by Birkhold in which interstitial Pb^{2+} is reduced to $Pb(0)$.⁶ Both I_i and interstitial $Pb(0)$ will be the recombination centers that lead to the dark area in the PL. Overall, the positive electrode will be enriched of iodine atoms as supported by XPS measurements carried out previously by our group.¹⁴ Alternatively, the negative electrode will be enriched with vacancies and when these are confined at the electrode can lead to formation of PbI_2 as described by Huang *et al.*¹⁵ in agreement with a Pb/I ratio lower than 3, also supported by XPS.¹⁴

It can be observed in different images that there is a bright edge close to the contacts indicating that the actual species present at the contacts are different to those present in the bulk with less concentration of PL quenching centers close to the contacts. As can be observed in the video S11 (from 8 seconds onwards) at the edge of the advancing front there is an increase in PL which could be the result of neutralization of I_i recombination centers by V_I , just the opposite to Frenkel defect generation. We propose that at the bright edge there is a large concentration of V_I susceptible to be transported under an applied electrical field as dominant moving species as proposed by Senocrate *et al.*⁵ and

low concentration of I_i as quenching species.

Here it is important to highlight that the direction of the advancing front will be highly dependent on the actual recombination species present in the films. Our results clearly show an advance of the dark front from the positive to the negative electrodes in agreement with different authors.^{4,15} Alternatively, there are other authors that report that the dark front advances from negative to positive.^{6,16,17} Indeed, it has been reported that the doping nature of the MAPbI₃ will strongly depend on the processing conditions to prepare the films and imbalance of the reagents with excess of MA⁺ or PbI₂ can lead to n-doping or p-doping, respectively.⁷

Fig.: Diagrams with a proposal of the processes taking place during the application of an electrical field in the lateral electrode configuration. a) and d) Represents the initial state with a background of p-dopants (P_0) and iodine vacancies present at the interface with the electrode. b) and e) When the electrical field is applied vacancies migrate towards the negative electrode, in this transit they partially compensate the charge of the background doping, including some non-radiative recombination defects (I_i). c and f) Excess vacancies carry excess of electrons that can lead to generation of other PL quenching defects such as Pb (0).

References

- 1 Azpiroz, J. M., Mosconi, E., Bisquert, J. & De Angelis, F. Defect migration in methylammonium lead iodide and its role in perovskite solar cell operation. *Energy Environ. Sci.* **8**, 2118-2127, (2015).
- 2 Yongbo, Y. *et al.* Electric-Field-Driven Reversible Conversion Between Methylammonium Lead Triiodide Perovskites and Lead Iodide at Elevated Temperatures. *Adv. Energy Mater.* **6**, 1501803, (2016).
- 3 Chen, S. *et al.* Mobile Ion Induced Slow Carrier Dynamics in Organic-Inorganic Perovskite CH₃NH₃PbBr₃. *ACS Applied Materials & Interfaces* **8**, 5351-5357, (2016).
- 4 Deng, X. *et al.* Electric field induced reversible and irreversible photoluminescence responses in methylammonium lead iodide perovskite. *J. Mat. Chem. C* **4**, 9060-9068, (2016).
- 5 Senocrate, A. *et al.* The Nature of Ion Conduction in Methylammonium Lead Iodide: A Multimethod Approach. *Angew. Chem. Int. Ed.* **56**, 7755-7759, (2017).
- 6 Bandiello, E. *et al.* Influence of mobile ions on the electroluminescence characteristics of methylammonium lead iodide perovskite diodes. *J. Mat. Chem. A* **4**, 18614-18620, (2016).
- 7 Birkhold, S. T. *et al.* Interplay of Mobile Ions and Injected Carriers Creates Recombination Centers in Metal Halide Perovskites under Bias. *ACS Energy Lett.* **3**, 1279-1286, (2018).
- 8 Wang, Q. *et al.* Qualifying composition dependent p and n self-doping in CH₃NH₃PbI₃. *App. Phys. Lett.* **105**, 163508, (2014).
- 9 Zhang, H. *et al.* Dynamic interface charge governing the current-voltage hysteresis in perovskite solar cells. *Phys. Chem. Chem. Phys.* **17**, 9613-9618, (2015).
- 10 Heiser, T. & Weber, E. R. Transient ion-drift-induced capacitance signals in semiconductors. *Phys. Rev. B* **58**, 3893-3903, (1998).
- 11 Sun, Q. *et al.* Role of Microstructure in Oxygen Induced Photodegradation of Methylammonium Lead Triiodide Perovskite Films. *Adv. Energy Mater.* **7**, 1700977, (2017).
- 12 Meggiolaro, D., Mosconi, E. & De Angelis, F. Modeling the Interaction of Molecular Iodine with MAPbI₃: A Probe of Lead-Halide Perovskites Defect Chemistry. *ACS Energy Lett.* **3**, 447-451, (2018).
- 13 Li, C. *et al.* Iodine Migration and its Effect on Hysteresis in Perovskite Solar Cells. *Advanced Materials* **28**, 2446-2454, (2016).
- 14 Yuan, Y. *et al.* Electric-Field-Driven Reversible Conversion Between Methylammonium Lead Triiodide Perovskites and Lead Iodide at Elevated

Temperatures. *Adv. Energy Mater.* **6**, 1501803, (2016).

15 Zhang, Y. *et al.* Reversible Structural Swell–Shrink and Recoverable Optical Properties in Hybrid Inorganic–Organic Perovskite. *ACS Nano* **10**, 7031-7038, (2016).

16 Jacobs, D. L., Scarpulla, M. A., Wang, C., Bunes, B. R. & Zang, L. Voltage-Induced Transients in Methylammonium Lead Triiodide Probed by Dynamic Photoluminescence Spectroscopy. *J. Phys. Chem. C* **120**, 7893-7902, (2016).

Response to reviewers comments

Reviewer #1

Using optical microscopy Li et al observed bias electric field induced PL on/off phenomenon in halide perovskite planar film with lateral symmetric electrodes. They confirmed that a dark region advances from the positive electrode at a slow velocity of order of $1\mu\text{m s}^{-1}$. They intended to address the sharp front moving in terms of the drift of ionic vacancies that drastically reduce the PL efficiency. They proposed a dynamic transport model and showed that the square reciprocal of the electrical current is linear with time.

Basically, this topic can be interesting because the observed phenomena are not yet fully or quantitatively understood to date. However, relative to the previous publications, the current manuscript does not provide significantly advance; and there is significant problem in this manuscript. To be considered for publication in Nature Communications, I think authors must carefully address the following concerns and significantly improve the manuscript:

Thank you for a careful reading of our manuscript. We think this new version and the general statement above have made clear the significant novelty and new insight supported by the analysis of experimental data.

1. Authors ignored the similar phenomena previously observed by other groups, for example, Deng et al. Electric field induced reversible and irreversible photoluminescence responses in methylammonium lead iodide perovskite, JMC-C 4(2016) 9060; Yuan et al. Electric Field Driven Reversible Conversion Between Methylammonium Lead Triiodide Perovskites and Lead Iodide at Elevated Temperatures, Advanced Energy Materials 6 (2), 2015.

Response: We thank the referee for bringing these articles to our attention that we unfortunately were unnoticed. This was a failure of coordination in the paper writing stage, not that we had the intention to ignore previous significant work. We have incorporated the two references in the current manuscript, and also we have made a significant revision of references and we think nothing important is left. On page 4, “These studies concluded that iodide vacancies are the dominant ionic species that moves under the influence of electrical field, in combination with an electronic current that flows through the sample. Huang and coworkers¹⁹ observed a moving thread in MAPbI₃. Ho-Bailie and coworkers reported the growing regions of quenched PL dependence on the size of electrical field and humidity.”

“Our results clearly show an advance of the dark front from the positive to the negative electrodes in agreement with different authors.^{11,19}”

“the negative electrode will be enriched with vacancies and when these are confined at the electrode can lead to formation of PbI₂ as described by Huang et al. ¹⁹”

2. “ionic vacancies drastically reduce the PL efficiency”? Authors need to make deep insight for the mechanism for ion induced PL quenching, although some previous publications, for example, Wen et al. Mobile charge-induced fluorescence intermittency in methylammonium lead bromide perovskite, Nano letters 15 (2015), 4644

Response: We have added the reference in main text. Concerning the deep insight for ion induced PL quenching, we have put the detailed discussion on interpretation of migrating species and its relationship with PL quenching in the final part of the paper, as depicted in Figure 4.

3. In experiment, the PL image is essential data. It is important to use the consistent scale bar for PL intensity, in addition the size. Authors should clearly interpret the relation of PL intensity variation—spatial—time.

Response: Both the size bar and PL intensity bar for all the images are consistent with all images, the scale bar represents 100 μm . We have added the PL intensity bar in Fig.1. The color bar is grey value with arbitrary unit, the same as used previously. (Nature 523, 196(2015)). The additional interpretation of the PL intensity variation against spatial and time is included in Figure S1 and Figure S6.

4. Illumination effect cannot be ignored, many papers already mention the mobile ion activation by the illumination, in addition to the electric field. The detailed illumination condition should be presented, such as during, pulse or continuous.

Response: We totally agree with the reviewer. The illumination condition has significant influence on the ion migration, for instance (Phys.Chem.Chem.Phys.18, 30484(2016))(Nat. Commun. 7, 11683(2016)). Besides, the external illumination generates charge carrier, therefore, the intensity of illumination determines the density of the charge carriers in the semiconductors. In this paper, the illumination condition is that “the excitation intensity was $\sim 35 \text{ mW/cm}^2$ with wavelength of 440 nm” and during the experiment, the illumination was kept as the constant intensity, in any case due to the low excitation intensities blinking was not observed. The description of the illumination condition was included in experimental section and supporting information, Figure S7. Concerning the excited illumination intensity dependent ion migration will be further addressed in upcoming work and would be beyond the focus of this manuscript.

5. Some statement lack evidence or justification, for example, “These findings were described in terms of a simple mechanism common to both fully inorganic and organic semiconductors (6, 7) in which photogenerated electrons and holes drift to opposite sides of the device, reducing the bulk recombination rate and hence PL intensity.” and “However, diffusion of the vacancies would cause a gradually decaying spatial distribution of PL rather than an advancing edge.”

Response: Concerning the statement: “These findings were described in terms of a simple mechanism common to both fully inorganic and organic semiconductors (6, 7) in which photogenerated electrons and holes drift to opposite sides of the device, reducing the bulk recombination rate and hence PL intensity.” This statement is within the introduction part, and we just summarize the previous work of other groups, in which a very simple model was used that doesn’t show any advancing front, maybe due to low field intensity. It indeed shows, that more research is required to fully understand the recombination i.e. PL dynamics in hybrid perovskites. Our manuscript will indeed provide a simple model that ascribes the PL quenching to the change in electron hole densities as a consequence of ion / vacancy migration.

Concerning the statement, “However, diffusion of the vacancies would cause a gradually decaying spatial distribution of PL rather than an advancing edge.”, as shown in Ref [Nat. Commun. 7, 11683(2016)], due to the illumination, there is a photoinduced ion redistribution. Since the ion gradient, there is a distribution of PL observed by confocal microscopy, shown in that paper. We have added the reference and interpretation into the main text. “Diffusion of ionic species ascribed to the ion concentration gradient, makes one assume a gradually decaying spatial distribution of PL [Nat. Commun. 7, 11683(2016)] rather than an advancing edge towards opposite electrode”. The fact is, diffusion alone can never explain that you get a sharp front. It is necessary to have ion drift. But as discussed in the paper, the advance of the front cannot be described only in terms of drifting ions, since the diffusion coefficient thus calculated is too low. Hence the need for our more advanced model.

6. Can authors provide a few simulations according the modelling? and explain the sharp edge shift.

Response: We agree with the suggestion from reviewer that it would be very interesting to obtain such results by simulation. The way to do it is to make a dynamic drift-diffusion simulator, with appropriate boundary conditions, and introduce assumptions about solid-state reactions. We do not have such tools but we are working on it. At the moment we have used some general assumptions and solved the transport equations. The output of the model is the position of the front and total current as a function of time. These results have been given in the paper analytically, and they describe very well the experimental observations.

Reviewer #2 :

While the selected type of experiment described here is interesting and able to reveal much information on the materials, it has to be pointed out that this work suffers from a variety of shortcomings:

1) The first point refers to novelty and originality. The authors essentially refer to experiments (with a slightly different composition) and qualitatively similar results that they already previously published in [Li C., Guerrero A., Zhong Y., Gräser A., Luna C. A. M., Köhler J., Bisquert J., Hildner R. and Huettner S., Real-Time Observation of Iodide Ion Migration in Methylammonium Lead Halide Perovskites. *Small* 13, 1–10 (2017)]. Although they couple these with a dynamic transport model, added value of this paper is not sufficient to justify publication in Nature Communication.

Response: In the current revision of the manuscript we have expanded the type of measurements to provide further insights and extended to different types of perovskites, as described above. In this paper we wish to explain quantitatively the observation of a moving dark area when a laterally contacted sample is biased under light soaking. We develop a quantitative model based on the idea that the current across the film) is predominantly electronic. This topic should be interesting to a broad audience because

the observed phenomena are not yet fully or quantitatively understood to this date.

2) Notwithstanding the fact that the authors focus on the effect of anion vacancies on the PL response of MAPbI₃, they strongly suggest that iodine interstitials are present in large number in the MAPbI₃ structure (generated by a Frenkel defect reaction). This claim is not supported by any evidence, especially considering that normally perovskites are dominated by Schottky defect reactions (metal and anion vacancies). The authors feel supported by recent measurements (Ref. 17 - Minns, J. L., Zajdel, P., Chernyshov, D., van Beek, W. & Green, M. A., Nat. Commun. 8, 15152 (2017)). However, in that report such interstitials are neutral (and therefore of much different size than negatively charged iodide ions) and form molecular aggregates. Such defects would not contribute to p-type conductivity, as claimed by the authors, but rather would represent a loss in electron holes, opposite to their claims and observations. On the same lines, the authors neglect the significant amount of literature calculating low formation energies for Schottky defects (for example, [1] Eames, C. et al., Nat. Commun. 6, 7497 (2015) -- [2] Haruyama, J., Sodeyama, K., Han, L. & Tateyama, Y., J. Am. Chem. Soc. 137, 10048–10051 (2015) –[3] Walsh, A., Scanlon, D. O., Chen, S., Gong, X. G. & Wei, S.-H., Angew. Chem. Int. Ed. 54, 1791–1794 (2015)).

Response: What kind of defects are prevalent is not necessarily overall important for the explanation and the model we provide. The reviewer is absolutely right, that Schottky defects show small formation energies and that they may be prevalent. But, very recent studies from 2018 give a more concise picture of the overall presence of Frenkel defects. One example are the experiments by the Ginger group, who use a very similar device geometry as we do, but include an insulating layer between the contacts and probe the charge density redistribution by Kelvin probe microscopy across the channel. Their experiments clearly support the prevalence of Frenkel defects and a long-range motion thereof. We have refined the manuscript to explain this fact in a better way and we have added various older and very recent references on this area.

3) The authors model the current behaviour of their samples by assuming the ionic current to be negligible. Simply by looking at Fig. 1c, this is evidently an unwarranted claim. Instead, it appears very clear how the ionic current (which is expected to decrease as a function of time due to progressive ion blocking) makes up the largest part of the total current during the initial biasing. Indeed, the value starts at 7 μA (ionic and electronic), while at 120 seconds it is decreased to 2 μA (only electronic). Such a claim could only be valid at long times, where the ionic current disappears, but cannot describe the development of the bias-induced polarization.

Response: We have commented above (and in the main text), the comparison of the two models.

It has been shown that the darkening area requires a large applied field and we believe in these conditions Hebb-Wagner classical polarization method does not describe well the experiment. In contrast to this, our model of local saturation of ion-induced doping modification implies that the square reciprocal of the electrical current is linear with time in agreement with the experimental observations. We are assuming that the dominant current is electronic and that ions disturb the local charge compensation while ionic current is negligible, except in a short initial instant. We have provided evidences above that a simply decaying ionic/electronic current does not explain the results.

According to our view that current starts at 7 μA (ionic and electronic), and decreases to 2 μA is very simply explained by the decrease of electronic carriers by a factor 1/3 due to the spread of defects.

4) The authors refer to ion transport in halide perovskite, but it appears they do not consider the full implications of such transport on these materials. Specifically, the authors only discuss ion accumulation at the interfaces, rather than considering that such polarization must affect the entire bulk of the material via effectively neutral stoichiometric variation, which is a basic phenomenon in solid state electrochemistry (example references are: [1] Wagner, C., *Ber. Bunsenges. Phys. Chem.* 60, 4–7 (1956). - [2] Hebb, M. H., *J. Chem. Phys.* 20, (1952). -- [3] Yokota, I., *J. Phys. Soc. Japan* 16, 2213–2223 (1961).).

Response: The method of stoichiometric (galvanostatic) polarization, assumes that the initial current is mixed electronic and ionic, while the final current is only electronic. We do not dispute the validity of previous results, but we consider that galvanostatic polarization is usually performed^{5,8,9} at lower current density that may not induce the moving dark front. In the other extreme, larger applied electrical field may damage the MAPbI_3 sample.^{4,18} Our model does consider the ionic distribution in the bulk perovskite, as shown in Fig. 4, and we believe that here Hepp-Wagner polarization conditions do not apply. In fact Maier has pointed out that applicability of Hepp-Wagner polarization requires a number of conditions to be satisfied.¹⁹ The model based on drift-diffusion and a saturation hypothesis, is simple on the one hand, but describes the measured currents very well, by being completely consistent with a second method – the PL microscopy experiments showing the advancing dark front on the other hand. The model includes the dynamic alternation of charge carrier densities within the material. In the future it may be very interesting to compare the results of both methods as a function of applied field, but this extensive investigation goes far beyond our current possibilities.

In summary, it appears that the results presented here suffer from several major scientific issues and apart from that do not provide sufficient advancement on the

present knowledge or enough novelty. I therefore cannot recommend publication in Nature Communications.

Response: We hope that we could clarify the pointed out scientific issues with the new data we provided (i.e. EIS, Seebeck coefficient, accounting for capacitive contributions, updating latest references on Frenkel defects). And we expect the new extended experimental analysis will convince the referee of the novelty and significance of our results. Please check in Fig. S10 the excellent description of the current transient in an array of different samples. When these results are treated quantitatively we are able to obtain the diffusion coefficient in agreement with Maier et al results by other methods. The implications should not be underestimated: we provide a consistent model which related electronic and optical properties of a metal organic perovskite and explain the role of vacancies including a way to estimate their density. Understanding and controlling defect by for example doping or passivation is one of the latest big research topics in perovskite photovoltaics community and increasingly engaged by many groups.

Reviewer #3:

In this manuscript, Cheng Li et al. report on the electrical switching of photoluminescence in lead halide perovskite and explain the underlying mechanism based on a simple dynamic transport model. In brief, the authors attribute the existence of the sharp front and movement of the sharp front of the luminescence to the drift of the ionic vacancies, which would reduce the doping and create the nonradiative recombination center leading to the reduced luminescence. They further extracted the ion diffusion coefficient based on the simple model they developed. Overall, the topic is interesting and important; however, the model is oversimplified by neglecting too much factors such as built-in electric field and experimental results cannot fully support their conclusions. Therefore, the referee cannot recommend its publication before the authors fully address the following questions.

1 The description is too simple and not clear at all for Figure 1. The authors are suggested to indicate the sharp front in Figure 1a and explain why the luminescence at 30s and 55s is so weak. If possible, it would be better to provide a video to show the movement of the sharp front and the change of the luminescence intensity.

Response: A video is now included and additionally we enhanced its data presentation (see updated Figure 1 and Figure 2). Furthermore we presented the detailed mechanism in Fig. 4.

2 The authors claimed that the recovery of ionic vacancies occurs at a much slower

rate after removing the bias voltage or basing in the contrary direction. How slow is this process? How does the current evolve after removing the bias? Can the authors provide more information on this?

We applied 9 V across the 1 mm channel for 5 seconds and measured the open circuit voltage as the function of time. This curve can be fitted by a single exponential function with time constant of ~ 1.5 second. This can provide an estimation of the ionic contribution decay. In Figure 2(a), we applied the contrary voltage and monitored the change of the current. The result indicates that the electric current plays an important role in the measurement.

3 The model the authors developed is based on the assumption that MAPbI₃ is p-doped at room temperature. Nevertheless, the authors did not provide any evidence. As a matter of fact, previous transport studies have demonstrated that MAPbI₃ at room temperature is n-doped (see Nature Communications, 7, 11330, 2016; Nature Communications, 6, 7383, 2015). Can the authors provide transport measurement to prove that the films they used are p-doped?

Response: We performed Seebeck coefficient measurements on perovskite thin films and with a lateral electrode geometry. They clearly show the p-type behaviour of the herein prepared samples. Please, note that these measurements belong to one of the first Seebeck coefficient measurements which have been carried out on organometal halide perovskites thin film. This will definitely generate a lot of attention within the community.

4 The authors attributed the decrease of the current and luminescence intensity to the drift of the ionic vacancies, which would reduce the doping and create the nonradiative recombination center leading to the reduced luminescence. Nevertheless, they did not provide any information on the concentration of the iodine vacancies and iodine interstitials. The decrease of the measured current is on the order of μA . Can the ionic drift reduce the hole concentration on such large degree?

Response: We thank the referee for this important question. We see in the current transient of Fig 1 that the current is changing from 12 to 3 mA cm⁻². In terms of the change of doping density this is only a factor 4 that can be easily induced by the mechanism of moving ions that change doping. Please note that our method, based on the variation of current density, only determines the relative change of doping, Eq. (16)

$$p_1 = \frac{1}{1+\gamma} p_0 = 0.23 p_0$$

and not the absolute value of doping. Furthermore we performed Seebeck coefficient measurements (as Shown in Figure S7) on thin films after biasing and obtained a value much larger than the initial one, from 3.4±0.5 mV/K to 6.2±0.8 mV/K, which clearly points to the significant changes of the electron hole densities. Furthermore, we would like to cite the latest paper by Birkhold et. al, who obtained charge carrier density variations of up to 2 x 10¹⁵ cm⁻³. (J. Phys. Chem C 2018, 122, 12633)

5 In the model the authors developed, they assume the electric field is uniform cross the region 1 and region 2 and neglect the built-in field due to the uneven distributed ions. Can the author justify this? The built-in field due to the ions is expected to be quite large if the concentration of ions is such large that can reduce the doping concentration so much.

Response: We are assuming ohmic transport throughout the layer, in the two regions. There is charge compensation everywhere and no space charge, so that the electric field can be considered uniform, but it changes between the two regions as they hold the same current with different carrier densities. This assumption will fail at the boundaries, including the boundary from dark to light region. However we believe the description of drift currents based on the model is accurate.

6 Can the authors provide PL spectra simultaneously with PL images shown in Figure 1a? The evolution of the peaks of PL spectra might be able to show useful information to support the authors assumption.

We have added the PL spectra in Figure 2(c), which shows that there is no shift of the PL peaks. This is consistent with our theory on the change of the defect state which influences the PL intensity.

7 The authors claimed that ‘When iodine vacancies move to the right they compensate the negative charge from iodine interstitials reducing the hole density as shown in Figure 1e.’ How can iodine vacancies compensate the negative charge from iodine interstitials? Do the interstitials capture the iodine vacancies? Can the authors illustrate more on this using schematic illustrations?

Response: We have provided a more complete discussion of this point in the above text and in the manuscript, please see Fig. 4.

8 The authors stated that the drift of iodine vacancies can create the nonradiative recombination sites. Can the authors explain how to achieve this?

Response: We have provided a more complete discussion of this point in the above text and in the manuscript, please see Fig. 4.

9 In equation (15), p_0 should be n_0 . In Figure 2, the change of the energy band would not expect to change so suddenly due to the existence of the build-in field.

Response: We have corrected the equation. We agree and also mention that the build-in field can compensate the field inside. However, it is the change of the doping of the materials that can significantly contribute the field distribution across the region 1 and 2. In the linear region, we assume the uneven distributed ions do not significantly change the field inside.

References

- 1 Azpiroz, J. M., Mosconi, E., Bisquert, J. & De Angelis, F. Defect migration in methylammonium lead iodide and its role in perovskite solar cell operation. *Energy Environ. Sci.* **8**, 2118-2127, doi:10.1039/c5ee01265a (2015).
- 2 Yongbo, Y. *et al.* Electric-Field-Driven Reversible Conversion Between Methylammonium Lead Triiodide Perovskites and Lead Iodide at Elevated Temperatures. *Adv. Energy Mater.* **6**, 1501803, doi:doi:10.1002/aenm.201501803 (2016).
- 3 Chen, S. *et al.* Mobile Ion Induced Slow Carrier Dynamics in Organic-Inorganic Perovskite CH₃NH₃PbBr₃. *ACS Applied Materials & Interfaces* **8**, 5351-5357, doi:10.1021/acsami.5b12376 (2016).
- 4 Deng, X. *et al.* Electric field induced reversible and irreversible photoluminescence responses in methylammonium lead iodide perovskite. *J. Mat. Chem. C* **4**, 9060-9068, doi:10.1039/c6tc03206k (2016).
- 5 Senocrate, A. *et al.* The Nature of Ion Conduction in Methylammonium Lead Iodide: A Multimethod Approach. *Angew. Chem. Int. Ed.* **56**, 7755-7759, doi:10.1002/anie.201701724 (2017).
- 6 Birkhold, S. T. *et al.* Interplay of Mobile Ions and Injected Carriers Creates Recombination Centers in Metal Halide Perovskites under Bias. *ACS Energy Lett.* **3**, 1279-1286, doi:10.1021/acseenergylett.8b00505 (2018).
- 7 Wang, Q. *et al.* Qualifying composition dependent p and n self-doping in CH₃NH₃PbI₃. *App. Phys. Lett.* **105**, 163508, doi:10.1063/1.4899051 (2014).
- 8 Kim, G. Y. *et al.* Large tunable photoeffect on ion conduction in halide perovskites and implications for photodecomposition. *Nat. Mater.*, doi:10.1038/s41563-018-0038-0 (2018).
- 9 Zhou, W. *et al.* Light-Independent Ionic Transport in Inorganic Perovskite and Ultrastable Cs-Based Perovskite Solar Cells. *J. Phys. Chem. Lett.* **8**, 4122-4128, doi:10.1021/acs.jpcclett.7b01851 (2017).
- 10 Zhang, H. *et al.* Dynamic interface charge governing the current-voltage hysteresis in perovskite solar cells. *Phys. Chem. Chem. Phys.* **17**, 9613-9618, doi:10.1039/c5cp00416k (2015).
- 11 Heiser, T. & Weber, E. R. Transient ion-drift-induced capacitance signals in semiconductors. *Phys. Rev. B* **58**, 3893-3903, doi:10.1103/PhysRevB.58.3893 (1998).
- 12 Sun, Q. *et al.* Role of Microstructure in Oxygen Induced Photodegradation of Methylammonium Lead Triiodide Perovskite Films. *Adv. Energy Mater.* **7**, 1700977, doi:doi:10.1002/aenm.201700977 (2017).
- 13 Meggiolaro, D., Mosconi, E. & De Angelis, F. Modeling the Interaction of

Molecular Iodine with MAPbI₃: A Probe of Lead-Halide Perovskites Defect Chemistry. *ACS Energy Lett.* **3**, 447-451, doi:10.1021/acseenergylett.7b01244 (2018).

14 Li, C. *et al.* Iodine Migration and its Effect on Hysteresis in Perovskite Solar Cells. *Advanced Materials* **28**, 2446-2454, doi:10.1002/adma.201503832 (2016).

15 Yuan, Y. *et al.* Electric-Field-Driven Reversible Conversion Between Methylammonium Lead Triiodide Perovskites and Lead Iodide at Elevated Temperatures. *Adv. Energy Mater.* **6**, 1501803, doi:10.1002/aenm.201501803 (2016).

16 Zhang, Y. *et al.* Reversible Structural Swell–Shrink and Recoverable Optical Properties in Hybrid Inorganic–Organic Perovskite. *ACS Nano* **10**, 7031-7038, doi:10.1021/acsnano.6b03104 (2016).

17 Jacobs, D. L., Scarpulla, M. A., Wang, C., Bunes, B. R. & Zang, L. Voltage-Induced Transients in Methylammonium Lead Triiodide Probed by Dynamic Photoluminescence Spectroscopy. *J. Phys. Chem. C* **120**, 7893-7902, doi:10.1021/acs.jpcc.5b11973 (2016).

18 Zhao, Y.-C. *et al.* Quantification of light-enhanced ionic transport in lead iodide perovskite thin films and its solar cell applications. *Light: Science & Applications* **6**, e16243, doi:10.1038/lsa.2016.243

<https://www.nature.com/articles/lsa2016243#supplementary-information> (2017).

19 Denk, I., Münch, W. & Maier, J. Partial Conductivities in SrTiO₃: Bulk Polarization Experiments, Oxygen Concentration Cell Measurements, and Defect-Chemical Modeling. *Journal of the American Ceramic Society* **78**, 3265-3272, doi:doi:10.1111/j.1151-2916.1995.tb07963.x (1995).

Reviewers' comments:

Reviewer #1 (Remarks to the Author):

In this revised manuscript, authors have significantly enhanced the experimental results and interpretation. The results for different samples and under different bias conditions are presented. The new measurement, IS and Seebach, are added; and also video for the moving. I think this manuscript has been significantly improved. Authors need to address some concerns before it can be published in Nature Communications:

1. Clear captions should be presented for each figure and video (including SI), for example, sample composition and bias conditions;
2. This work shows large different diffusion coefficient with respect to previous work, it is necessary clearly present the sample details and basic characterization for structure and stoichiometric of samples;
3. In terms of the interpretation for PL quenching, what role does CI play? taking into account significant CI in the sample.

Reviewer #2 (Remarks to the Author):

Compared with the previous version, the authors have added more examples and worked on the interpretation. Yet, I do not think the interpretation is correct. It suffices to refer to two basic points.

- 1) The authors state that the diffusion front occurring at high bias (not seen at lower bias) is indicative of an effect not due to stoichiometric polarization. Yet, it is known in the literature that such fronts exactly occur as a consequence of stoichiometric polarization. At low bias the profile is smooth, but at high bias, the chemical diffusion coefficient that contains concentration terms can no longer be considered as a constant and steep front-like profiles occur (see literature on titanates); exactly as the authors find.
- 2) The high frequency resistance changes as a function of voltage (Figure 3). The authors use this finding to discard stoichiometric polarization. Again, just the opposite is true: Exactly this picture is expected for a stoichiometric polarization (including Figure 3a) as the degree of stoichiometric polarization, and hence the mean conductivity, is a function of voltage, even in the steady state. In summary, the paper should not be published, as (i) the results are completely expected for mixed conductors under bias as a consequence of ion-blocking electrodes (cf. work on titanates in the literature) and (ii) are not correctly interpreted and publication would increase the already existing confusion in the literature.

Reviewer #3 (Remarks to the Author):

Although the manuscript has been significantly improved, more experimental results are required to test their proposed model and to support their conclusions. Only by doing so, I can recommend its publication.

- 1 If the moving of the dark front is indeed due to the ion migration, the authors should perform experiments on perovskite single crystals, when the ion migration is greatly suppressed. Alternatively, the authors can tune the ion concentration via tuning the synthesis procedure and see how this moving changes with ion concentration.
- 2 As predicated by the proposed model, the current should increase with time for n-type perovskites,

why did not the authors carry out the same experiments on n-type perovskites to exam this?

3 The applied electric field is around $4\text{V}/150\text{ }\mu\text{m}$ ($\sim 26.6\text{V}/\text{cm}$). I donot think that such small field can induce such large ion migration. The authors can check this by scanning IV loops and see whether this is hysteresis under such small field.

4 For 2D perovskites, previous studies suggest that no ion migration has been observed [see: ACS Energy Lett. 2017, 2, 1571–1572; ACS Energy Lett. 2018, 3, 684-688]. However, the moving of the dark front is still observed here in 2D perovskites. Can the authors explain this?

5 In terms of the decrease of the current, I cannot believe that barely migration of the ionic vacancies can cause such large current drop (in response of #4 of my comments). Is there any electronic current?

6 I still have problem to understand how the drift of ionic vacancies creates the nonradiation recombination center. The authors should provide direct evidences for this since this is indispensable to explain the moving of dark front due to the ion drift.

Reviewer #1 (Remarks to the Author):

In this revised manuscript, authors have significantly enhanced the experimental results and interpretation. The results for different samples and under different bias conditions are presented. The new measurement, IS and Seeback, are added; and also video for the moving. I think this manuscript has been significantly improved. Authors need to address some concerns before it can be published in Nature Communications:

1. Clear captions should be presented for each figure and video (including SI), for example, sample composition and bias conditions;

We have added the necessary information on all figures and videos.

2. This work shows large different diffusion coefficient with respect to previous work, it is necessary clearly present the sample details and basic characterization for structure and stoichiometric of samples;

We have added the structure and stoichiometric information, such as XRD and XPS data in Supporting information (Figure S11 and S12). We have summarized all parameters calculated in different samples (Table 1). We agree that the structure and stoichiometric of samples are strongly associated with measured diffusion coefficient. Although all the devices were fabricated in the same N₂ glovebox, the atmosphere (solvent left) in glovebox and glovebox ambient temperature during spinning coating, still have influence on the samples.

3. In terms of the interpretation for PL quenching, what role does Cl play? taking into account significant Cl in the sample.

The presence of Cl in perovskite has demonstrated the improved morphology and enhanced optoelectronic properties, such as increased diffusion length^[1] and longer carrier life time^[2] It is accepted that there is not significant Cl left in the sample during the fabrication. During the formation of MAPbI₃, MACl (gas) is released.^[3] The concentration of Cl in the film is much lower than the composition stoichiometry in the precursor solution.^[4] This is consistent with the intensity of Cl in our XPS data, which is approach the measurement limitation, as shown in Supporting Information.(Figure S11 and S12)

In terms of PL, we have compared with the pure MAPbI₃ and MAPbI_{3-x}Cl_x, which shows that MAPbI_{3-x}Cl_x shows higher PL intensity. This might be due to the improved morphology. In addition, the precursor of perovskite is MAI:PbCl₂=3:1, if there are MA⁺ left (or Cl left in the film), which usually lead to the more p-type semiconductor.^[5] Based on our theory, due to the higher density of defect, higher doping results in higher ion mobility and quicker PL quenching under the electric field.

Reviewer #2 (Remarks to the Author):

Compared with the previous version, the authors have added more examples and worked on the interpretation. Yet, I do not think the interpretation is correct. It suffices to refer to two basic points.

1) The authors state that the diffusion front occurring at high bias (not seen at lower bias) is indicative of an effect not due to stoichiometric polarization. Yet, it is known in the literature that such fronts exactly occur as a consequence of stoichiometric polarization. At low bias the profile is smooth, but at high bias, the chemical diffusion coefficient that contains concentration terms can no longer be considered as a constant and steep front-like profiles occur (see literature on titanates); exactly as the authors find.

2) The high frequency resistance changes as a function of voltage (Figure 3). The authors use this finding to discard stoichiometric polarization. Again, just the opposite is true: Exactly this picture is expected for a stoichiometric polarization (including Figure 3a) as the degree of stoichiometric polarization, and hence the mean conductivity, is a function of voltage, even in the steady state.

In summary, the paper should not be published, as (i) the results are completely expected for mixed conductors under bias as a consequence of ion-blocking electrodes (cf. work on titanates in the literature) and (ii) are not correctly interpreted and publication would increase the already existing confusion in the literature.

The experiment with an advance front of PL quenching has been reported in several publications by different groups, as described in our previous response and in the introduction to this paper. No authors so far have suggested a connection of this observation with the conditions of stoichiometric polarization. We are sorry that our efforts to demonstrate our point empirically did not find any explicit response by the referee. The referee has only presented generic statements of the type "it is known in the literature" or "cf. work on titanates in the literature" which could not be replied. We think the opinion of the referee is very personal. We believe if the referee is convinced of such explanation he or she should prepare their own paper explaining so, but our work uses new ideas strongly based in different types of evidences and should not be blocked, in the benefit of scientific advance.

Reviewer #3 (Remarks to the Author):

Although the manuscript has been significantly improved, more experimental results are required to test their proposed model and to support their conclusions. Only by doing so, I can recommend its publication.

1 If the moving of the dark front is indeed due to the ion migration, the authors should perform experiments on perovskite single crystals, when the ion migration is greatly suppressed. Alternatively, the authors can tune the ion concentration via tuning the synthesis procedure and see how this moving changes with ion concentration.

We agree we should carry out the same experiment in single crystal, however, we have not managed to grow enough size of single crystal for the PL imaging experiments. We do agree that it is necessary to observe the change of ion mobility by tuning the ion concentration. However, due to the complex formation process, we cannot control the ion concentration precisely. Recently, we are working with Prof. Nam Gyu Park's group by doping with different amount of K element and investigating the influence of doping.^[6] It is found the K doping can significantly suppress the ion migration by reducing the mobile ions.^[7] By calculation, K ions main occupy interstitial site, hence, both of the formation of iodide Frenkel defect and iodide

ion movement are significantly suppressed. Therefore, by doping K, the ion migration is significantly suppressed, as shown in Figure 1. However, when doping increases, above 15 μmol , the crystalline structure may not be stable. As a result, doping introduces more defects inside and increasing the ion mobility again, which is consistent with the J-V curve measurement in the paper. These results will be published in a separate paper with Park group.

Figure 1. Concentration of K dependence of ion mobility.

2 As predicated by the proposed model, the current should increase with time for n-type perovskites, why did not the authors carry out the same experiments on n-type perovskites to exam this?

Figure 2. a) Schematic diagram of PL imaging measurement under bias and the time dependent PL intensity across the channel of the film. b) Time dependent current under bias. It is noted that the current of this MAPbI₃ device increases with time under the bias. c) Time dependent current under different voltages.

We agree that it is necessary to carry out the same experiment both in n type and p type perovskite film. As discussed with Lukas Schmidt-Mende's group, they carried out the same experiment as shown in Fig 2.(a).^[8] As the precursor is MAI:PbI₂=1:1, they observed that the black front moving from negative towards positive. From them, we have obtained the time dependent current of their devices. In their devices, the current increases with time (Figure 2(b) and Figure 2(c)), and the moving direction is opposite with ours (Figure 2(a)). Based on our theory, this is consistent with our theory that n type semiconductor, the current increases with time and moving direction is from negative towards positive.

3 The applied electric field is around $4\text{V}/150\text{ }\mu\text{m}$ ($\sim 26.6\text{V}/\text{cm}$). I do not think that such small field can induce such large ion migration. The authors can check this by scanning IV loops and see whether this is hysteresis under such small field.

Figure 3. a) J-V curve of a $\text{MAPbI}_x\text{Cl}_{3-x}$ perovskite solar cell (FTO/ TiO_2 / $\text{MAPbI}_x\text{Cl}_{3-x}$ /Spiro-oMeTAD/Au) under AM 1.5 condition. b) J-V curve scanning at around V_{oc} , with small voltage $\sim 0.05\text{V}$.

At around V_{oc} , the internal field is considered as zero. We swept the voltage near V_{oc} with a small voltage ($\sim 0.05\text{ V}$ across 450 nm film). Figure 3(b) shows that this small voltage can drive ions resulting in the hysteresis in J-V curves. We have added the figures in supporting information.

4 For 2D perovskites, previous studies suggest that no ion migration has been observed [see: ACS Energy Lett. 2017, 2, 1571–1572; ACS Energy Lett. 2018, 3, 684–688]. However, the moving of the dark front is still observed here in 2D perovskites. Can the authors explain this?

Figure 4. a) GIWAXS characterization of a 2D multiple quantum wall (MQW) perovskite film on glass. b) Activation energy of ion migration in 2D multiple quantum wall perovskite film.

Considering 2D perovskite, in this work, we used 1-naphthylmethylamine iodide (NMAI), formamidinium iodide (FAI) and PbI_2 with a molar ratio of 2:1:2 dissolved in N,N-dimethylformamide (DMF) as the precursor to deposit perovskite films. The same film has been obtained in Wang, et al.^[9] In these films, rather than single value of n , there exists a range of n , forming the multipole quantum wall structure. As n becomes large, the property is more like 3D perovskite. As shown in Figure 4(a), apart from separated bright dots, there are also continuous rings in the GIWAXS pattern, indicating the random orientation. In addition, we also obtain the activation energy of iodide ions, which is around 0.5 eV , as shown in Figure 4(b). This value is much larger than the 3D perovskite, $0.2\text{--}0.3\text{ eV}$,^[10] implying larger barrier for ion migration in 2D MQW structures. In papers^[11], due to highly ordered film or single crystal, the authors did not obtain the activation energy for ion movement, so they claim that the ion migration is absent in 2D perovskite.

5 In terms of the decrease of the current, I cannot believe that barely migration of the ionic vacancies can cause such large current drop (in response of #4 of my comments). Is there any electronic current?

This is a central point of the discussion. Indeed we are claiming that the main current observed is electronic. However such electronic current is controlled by the number of defects. Under the bias, as we showed in thermoelectric Seebeck coefficient measurement, the doping level is significantly modified, from p type toward the intrinsic semiconductor. If defects change the doping by a fraction $\frac{1}{2}$, it follows that the electronic current changes by a factor $\frac{1}{2}$. Therefore the observed change of current is not so large when seen from the perspective of an electronic current

6 I still have problem to understand how the drift of ionic vacancies creates the nonradiation recombination center. The authors should provide direct evidences for this since this is indispensable to explain the moving of dark front due to the ion drift.

Figure 5 Temperature dependence of PLQY of different perovskite film.

Based on DFT calculation, interstitial iodide ion is one of the initial dopants that could be neutralized by the reverse of Frenkel type defect chemistry, this is the combination of vacancy and iodine interstice to eliminate the quenching site. Then, the excess iodine vacancies can lead to redox reactions as those described by Birkhold *et al.*^[8] in which interstitial Pb^{2+} is reduced to $Pb(0)$. Therefore, I_i and interstitial $Pb(0)$ will be the recombination centers that lead to the dark area in the PL. In our paper^[12], $Pb(0)$ signal has been observed in the channel.

In addition, there are also several references indicating that iodide vacancies are playing an important role in the nonradiative losses.^[13] Abdi-Jalebi *et al* found that KI is used to fill the halide vacancies, leading to the increase of PLQE to exceed 95%.^[7] Brenes *et al* also proposed that halide vacancies are the main source for shallow defect, which contributes to the nonradiative loss. These shallow defects can be passivated by light and O_2 .^[14]

We also carried out the experiments to compare the doping influence on the PL properties. As shown in Figure 5, $MAPbI_xCl_{3-x}$ is fabricated by the precursor (MAI:PbCl₂=3:1), which results in iodide interstitial defect. While $MAPbI_3$ is fabricated by the precursor (MAI:PbI=1:1), which results in more iodide vacancies defect. Figure indicates that $MAPbI_3$ demonstrates lower PLQE compared with $MAPbI_xCl_{3-x}$, implying that iodide vacancies act as the quencher for PL intensity. The temperature dependence of the PLQE is mainly due to the sharpening of the distribution function of charge carriers under low temperature.^[15]

References:

- [1] S. D. Stranks, G. E. Eperon, G. Grancini, C. Menelaou, M. J. P. Alcocer, T. Leijtens, L. M. Herz, A. Petrozza, H. J. Snaith, *Science* **2013**, *342*, 341.
- [2] S. T. Williams, F. Zuo, C.-C. Chueh, C.-Y. Liao, P.-W. Liang, A. K. Y. Jen, *ACS Nano* **2014**, *8*, 10640; Q. Chen, H. Zhou, Y. Fang, A. Z. Stieg, T.-B. Song, H.-H. Wang, X. Xu, Y. Liu, S. Lu, J. You, P. Sun, J. McKay, M. S. Goorsky, Y. Yang, *Nat. Commun.* **2015**, *6*, 7269.
- [3] H. Yu, F. Wang, F. Xie, W. Li, J. Chen, N. Zhao, *Adv. Funct. Mater.* **2014**, *24*, 7102.
- [4] J. You, Z. Hong, Y. Yang, Q. Chen, M. Cai, T.-B. Song, C.-C. Chen, S. Lu, Y. Liu, H. Zhou, Y. Yang, *ACS Nano* **2014**, *8*, 1674.
- [5] Q. Wang, Y. Shao, H. Xie, L. Lyu, X. Liu, Y. Gao, J. Huang, *Appl. Phys. Lett.* **2014**, *105*, 163508.
- [6] D.-Y. Son, S.-G. Kim, J.-Y. Seo, S.-H. Lee, H. Shin, D. Lee, N.-G. Park, *J. Am. Chem. Soc.* **2018**, *140*, 1358.
- [7] M. Abdi-Jalebi, Z. Andaji-Garmaroudi, S. Cacovich, C. Stavrakas, B. Philippe, J. M. Richter, M. Alsari, E. P. Booker, E. M. Hutter, A. J. Pearson, S. Lilliu, T. J. Savenije, H. Rensmo, G. Divitini, C. Ducati, R. H. Friend, S. D. Stranks, *Nature* **2018**, *555*, 497.
- [8] S. T. Birkhold, J. T. Precht, H. Liu, R. Giridharagopal, G. E. Eperon, L. Schmidt-Mende, X. Li, D. S. Ginger, *ACS Energy Lett.* **2018**, *3*, 1279.
- [9] N. Wang, L. Cheng, R. Ge, S. Zhang, Y. Miao, W. Zou, C. Yi, Y. Sun, Y. Cao, R. Yang, Y. Wei, Q. Guo, Y. Ke, M. Yu, Y. Jin, Y. Liu, Q. Ding, D. Di, L. Yang, G. Xing, H. Tian, C. Jin, F. Gao, R. H. Friend, J. Wang, W. Huang, *Nat. Photon.* **2016**, *10*, 699; Y. Wei, M. Li, R. Li, L. Zhang, R. Yang, W. Zou, Y. Cao, M. Xu, C. Yi, N. Wang, J. Wang, W. Huang, *Appl. Phys. Lett.* **2018**, *113*, 041103; J. Chang, S. Zhang, N. Wang, Y. Sun, Y. Wei, R. Li, C. Yi, J. Wang, W. Huang, *J. Phys. Chem. Lett.* **2018**, *9*, 881.
- [10] C. Li, A. Guerrero, Y. Zhong, S. Huettner, *Phys.: Condens. Matter* **2017**, *29*, 193001.
- [11] Y. Lin, Y. Bai, Y. Fang, Q. Wang, Y. Deng, J. Huang, *ACS Energy Lett.* **2017**, *2*, 1571; X. Xiao, J. Dai, Y. Fang, J. Zhao, X. Zheng, S. Tang, P. N. Rudd, X. C. Zeng, J. Huang, *ACS Energy Lett.* **2018**, *3*, 684.
- [12] C. Li, S. Tscheuschner, F. Paulus, P. E. Hopkinson, J. Kiebling, A. Köhler, Y. Vaynzof, S. Huettner, *Adv. Mater.* **2016**, *28*, 2446.
- [13] S. D. Stranks, R. L. Z. Hoyer, D. Di, R. H. Friend, F. Deschler, *Adv. Mater.* **2018**, DOI:10.1002/adma.201803336.
- [14] R. Brenes, D. Guo, A. Osherov, N. K. Noel, C. Eames, E. M. Hutter, S. K. Pathak, F. Niroui, R. H. Friend, M. S. Islam, H. J. Snaith, V. Bulović, T. J. Savenije, S. D. Stranks, *Joule* **2017**, *1*, 155.
- [15] C. L. Davies, M. R. Filip, J. B. Patel, T. W. Crothers, C. Verdi, A. D. Wright, R. L. Milot, F. Giustino, M. B. Johnston, L. M. Herz, *Nat. Commun.* **2018**, *9*, 293.

EVIEWERS' COMMENTS:

Reviewer #1 (Remarks to the Author):

I think authors have reasonably addressed referees' concerns and questions. To date, the reported behavior of mobile ions in perovskites is not really consistent, mostly likely due to the details of the materials. One important factor is that the halide ion/vacancy can be generated by electric field or illumination, and then depletion when keeping in dark, in addition to the existing amount of defects during fabrication. This revised manuscript provides a specific case, with a proposed interpretation, as a reference for the community. I recommend publishing this manuscript.

Reviewer #3 (Remarks to the Author):

Authors have addressed my concerns appropriately. Thus, the current version of this manuscript is recommended to be published.